# Sperm hyperactivation drives a circling-and-wandering swimming behavior

Meisam Zaferani [1,2,3] ✉, Yanis Baouche [4], Yamilka Lago-Alvarez[5], Anish Pandya[4], Soon Hon Cheong [5], Sabine Petry [2], Christina Kurzthaler [4,6,7] ✉ & Howard A. Stone [1] ✉

During migration through the female reproductive tract, sperm undergo physiological changes known as capacitation, including a motility transition termed hyperactivation. Hyperactivation is essential for various aspects of fertilization, particularly effective migration within the tract. However, how hyperactivation facilitates this migration remains elusive. Here, we profiled bull sperm hyperactivation and swimming in Newtonian and complex fluids, using microfluidic surfaces to mimic confinement of the tract. We identified three swim gaits: wandering (persistent random walks), circling, and an intriguing circling-and-wandering mode marked by stochastic transitions between the two. All gaits exhibit diffusive behavior over long time scales, with wandering showing a tenfold higher diffusivity than circling, and the effective diffusivity of circling-and-wandering falling in between. We found that while wandering sperm scatter from convex and concave surfaces, circling sperm become trapped around pillars, highlighting a distinctive feature of each phase. Additionally, stochastic simulations of active transport in porous media showed that as the geometrical complexity of the environment increases, circling-and-wandering outperforms either motility alone in spreading through the media. Our findings suggest that wandering may broaden the search landscape, while circling could help maintain local focus. Therefore, the combined circling-and-wandering swimming behavior might provide a flexible mechanism for modulating motility and facilitating migration in complex environments. Our results may have implications for understanding the physical aspects of sperm migration in the female reproductive tract.

For fertilization, mammalian sperm navigate the complex environment of the female reproductive tract (FRT) and withstand selective pressures. This migration is regulated by cues provided by the tract. Previous studies have shown that when swimming progressively in straight lines, sperm migrate upstream in response to external fluid flows[1–3]. Furthermore, physical boundaries, such as the walls of the FRT or microfluidic channels, direct the progressive motion of the sperm[4–7]. These observations suggest that the mucus flow and the architecture of the tract may modulate sperm migration, allowing them to traverse the lower regions of the tract and reach its upper part.

[1]Department of Mechanical and Aerospace Engineering, Princeton University, Princeton, NJ, USA. [2]Department of Molecular Biology, Princeton University, Princeton, NJ, USA. [3]Omenn-Darling Bioengineering Institute, Princeton University, Princeton, NJ, USA. [4]Max Planck Institute for the Physics of Complex Systems, Dresden, Germany. [5]Department of Clinical Sciences, College of Veterinary Medicine, Cornell University, Ithaca, NY, USA. [6]Center for Systems Biology Dresden, Dresden, Germany. [7]Cluster of Excellence Physics of Life, TU Dresden, Dresden, Germany. ✉e-mail: mzaferani@princeton.edu; ckurzthaler@pks.mpg.de; hastone@princeton.edu

During progressive swimming, sperm motility consists of a slightly asymmetric two-dimensional (2D) flagellar beating pattern[8–11], leading to helical motion in bulk fluid and circular motion near surfaces. Furthermore, sperm have been observed to roll unidirectionally around their longitudinal axis[12]. Previous observations on rodent sperm and computational analyses suggest that the slightly tilted head, with respect to the flagellar beating plane, causes this rolling motion[10,13–16].

However, as sperm enter the upper part of the tract, they become capacitated[17,18] and their flagellar beating patterns become highly asymmetric[19,20], shifting progressive motility to hyperactivated motility, which is characterized qualitatively by diverse movement patterns that trace erratic trajectories[18,21–23]. It is known that hyperactivation is caused primarily by the activation of the CatSper channel, which is distributed throughout the flagellum[24–26]. The structure and function of this channel[27–29], its evolutionary origin[30], and its necessity for fertilization[31,32] are well established. In addition, hyperactivation—induced by chemical[33,34] and/or thermal[35,36] stimuli—is known to facilitate sperm migration within the geometrically complex upper FRT, lined with heterogeneous and poorly characterized mucus[23,26,37–43]. However, a physical framework to elucidate how hyperactivation modulates sperm migration through rheological and geometrical complexities has yet to be developed.

Here, we show that hyperactivated motility manifests itself in distinct forms of near-surface motion depending on the rheological properties of the fluid. Combining microfluidic experiments, theoretical analysis, and stochastic simulations, we explore the implications of these sperm motility patterns for their interactions with surfaces and their motion through crowded environments, which represent important steps towards unraveling the physics of sperm migration within the FRT. This work refines the qualitative understanding of hyperactivation and lays the foundation for a physical model of mammalian sperm chemotaxis, a question that, unlike the well-studied chemotaxis in marine invertebrates[44–47], remains open.

## Results

### Statistical profiling of hyperactivated motility

We characterized the motility of bull sperm in an otherwise quiescent fluid in a circular microfluidic chamber (diameter = 1200 μm and height = 30 μm), designed to mimic the high-aspect-ratio geometry of the sperm passage in the FRT, such as microgrooves and narrow spaces between mucosal folds[41,48]. To induce sperm hyperactivation, 3.0 and 6.0 mM of caffeine, an established hyperactivation agonist[22,49,50], were added to the medium. We used Tyrode's albumin lactate pyruvate (TALP) as a standard Newtonian buffer, and supplemented it with 1% polyacrylamide (PAM, molecular weight 5–6 MDa) to reproduce the shear-thinning viscoelastic behavior of mucus. Detailed rheological characterization of this complex fluid and its comparison with bovine cervical mucus are provided in ref. 51. Sperm motion was then recorded at 15 frames per second (fps) for durations of 3–15 min.

**Progressive motion in a Newtonian fluid.** Bull sperm consist of a paddle-shaped head (of length $l_h = 5$ μm) and a flagellum whose wave-like bending induces swimming. This form of motility in TALP resulted in progressive motility, i.e., persistent movement on straight trajectories near the top and bottom surfaces of the chamber, at a translational speed of $101 \pm 11$ μm s$^{-1}$. Their swimming was subject to small noise, with a rotational diffusivity of $D_{rot} = 0.05 \pm 0.02$ s$^{-1}$. This noise arises from active swimming, rather than from typical Brownian motion. Observing at high temporal resolution (100 fps) and using the paddle-shaped head, we identified a nearly two-dimensional flagellar beating pattern (Fig. S1a). We also measured the rolling frequency[16,52] as $9.0 \pm 2.0$ Hz (Fig. 1a, b and Movie S1). The rolling is presumably induced by the tilt of the sperm head relative to the flagellar beating plane, at an angle of $\phi = 13.0° \pm 3.5°$ (Fig. S1b, c), which is consistent with the torque-free condition of microswimmers in viscous flows[13].

**Wandering motion in a Newtonian fluid.** When stimulated with hyperactivation agonists, the flagellar beating pattern becomes highly asymmetric, leading to stochastic wandering motion in TALP with features of a persistent random walk (PRW)[53]. Briefly, rolling counteracts the beat asymmetry, but the highly asymmetric beating patterns cause deviations from straight trajectories in random directions. The magnitude of these deviations depends on the level of asymmetry, so the intensity of hyperactivation determines the sperm's persistence length during wandering.

To quantify this wandering motion, we induced hyperactivation with 6.0 mM caffeine and extracted sperm trajectories over long durations (5–8 min, Fig. 1c and Movie S2). We then computed the mean-square displacement (MSD) $\langle |\Delta \vec{r}(t)|^2 \rangle$ from the trajectories with $\Delta \vec{r}(t) = \vec{r}(t) - \vec{r}(0)$ being the displacement between the initial $\vec{r}(0)$ and current $\vec{r}(t)$ center-of-mass positions. A representative example is shown in Fig. 1e, exhibiting short-time ballistic behavior ~$t^2$, followed by diffusive behavior $\langle |\Delta \vec{r}(t)|^2 \rangle = 4 D_{eff}^W t$, with an effective diffusivity of $D_{eff}^W = 1074 \pm 50$ μm$^2$ s$^{-1}$ at long times. Note that at very long times, the MSD saturates, reflecting the finite size of the microfluidic chamber. The diffusive regime is induced by sudden reorientations at randomly distributed times. We found that on average, the sperm swam at a net translational speed of $U = 20 \pm 4$ μm s$^{-1}$ and with a persistence time of $\tau = 4.2 \pm 2.2$ s ($N = 20$). Describing the motion as PRW in two dimensions, the diffusivity obeys $D_{eff}^W = U^2/(2\lambda)$, with $\lambda = D_{rot} + \tau^{-1}$. In agreement with previous reports[53], this wandering motion was dose-dependent, with $U = 80 \pm 27$ μm s$^{-1}$ and $\tau = 11.6 \pm 3.4$ s ($N = 20$) when stimulated with 3.0 mM caffeine. We further quantified wandering motion (6.0 mM caffeine) at the population level ($N = 392$ across 5 samples) and found a large distribution of translation speeds ranging from $U = \mathcal{O}(1)$ μm s$^{-1}$ to $U = \mathcal{O}(100)$ μm s$^{-1}$, which implies a large variation in $D_{eff}^W = \mathcal{O}(10) - \mathcal{O}(10^4)$ μm$^2$ s$^{-1}$ (Fig. 1f).

**Circling motion in complex fluids.** Upon entering a chamber filled with TALP + 1% PAM medium, a mucus-mimicking complex fluid, sperm rolling becomes suppressed and the cells follow circular trajectories with large radii, which stem from slight asymmetries in the beating patterns[41,53]. However, the addition of 6.0 mM caffeine results in highly asymmetric flagellar beating patterns, and the sperm exhibit circling with much smaller radii (Fig. 1e and Movie S2). Briefly, an asymmetric beating pattern generates local variations in force along the flagellum, which, while maintaining zero net force and torque, induce rotation during each beat cycle. This rotation gradually biases the swimming direction, producing a circular trajectory whose curvature increases with the level of asymmetry[8].

A typical MSD is shown in Fig. 1e and displays three distinct features: persistent ~$t^2$ behavior at short times, oscillatory dynamics at intermediate times—indicative of circling—and diffusive ~$t$ scaling at long times, with an effective diffusivity of $D_{eff}^C = 57 \pm 5$ μm$^2$s$^{-1}$. This circling motion has been studied theoretically, with predictions yielding an effective diffusivity[54,55] of $D_{eff}^C = U^2 D_{rot}/(2(D_{rot}^2 + \Omega^2))$, which decreases with increasing rotational speed ($\Omega$). We quantified sperm circling at the population level with ($N = 540$ across 5 samples) and without ($N = 199$ across 3 samples) treatment with 6.0 mM caffeine (Fig. 1f), and measured swimming parameters including $U$, $\Omega$, and $D_{eff}^C$. Caffeine treatment increases $\Omega$ from $0.4 \pm 0.3$ s$^{-1}$ to $1.8 \pm 0.8$ s$^{-1}$, which led to a substantial decrease of $D_{eff}^C$ from $(0.4 \pm 0.1) \times 10^3$ μm$^2$ s$^{-1}$ to $(0.2 \pm 0.1) \times 10^3$ μm$^2$ s$^{-1}$. We also found that the population-level average of $D_{eff}^W = (1.3 \pm 0.9) \times 10^3$ μm$^2$s$^{-1}$ is an order of magnitude higher than that for circling. Therefore, the transition from wandering in a Newtonian fluid to circling in a complex fluid significantly limits spatial exploration while enhancing local exploitation.

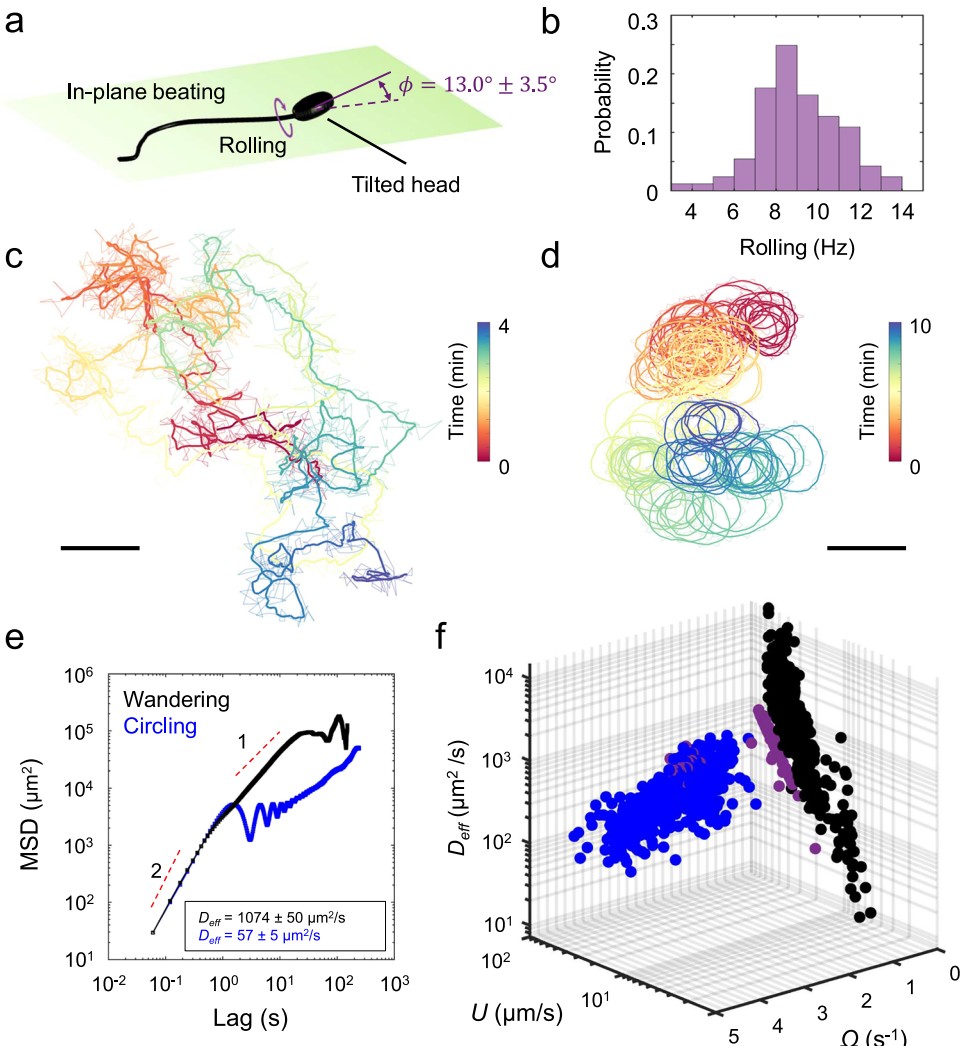

**Fig. 1 | Statistical profiling of hyperactivated motility. a** Schematic of bull sperm morphology and motility. The sperm head is slightly tilted out of the flagellar beating plane, causing rolling around the longitudinal axis. $\phi$ is the tilt angle of the sperm head with respect to the beating plane. **b** Rolling frequency measured at 100 fps for $N = 216$ sperm cells. **c** Representative trajectory of sperm wandering in TALP. Scale: 100 μm. **d** Circling motility in TALP + 1% PAM + 6.0 mM caffeine. These trajectories are recorded at 15 fps. Both real-time (transparent lines) and smoothed (solid lines) trajectories are shown. Scale: 100 μm. **e** MSDs (mean-square displacement) for wandering and circling motility as a function of lag time. **f** The swimming behavior space for wandering (black, $N = 392$) and circling (blue, $N = 540$) phases. Purple dots (as control, $N = 199$) represent data for circling sperm in TALP + 1% PAM without caffeine stimulation. $U$, $\Omega$, and $D_{eff}$ are the sperm's translational speed, rotational speed, and effective diffusion coefficient, respectively. Source data for **b**, **e** and **f** are provided as a Source Data file.

**Circling-and-wandering in complex fluids.** Under stimulation with 6.0 mM caffeine, suppression of rolling in TALP + 1% PAM medium was observed in 91 ± 5% of the cells (across 3 samples, Fig. S2a). However, suppression of rolling was not permanent for all sperm cells, and 27 ± 4% of them resumed rolling (Table S1). This subpopulation intermittently switched between non-rolling and rolling states, resulting in a mixed motility pattern alternating between circling and wandering periods. To accurately characterize this mixed motility and obtain reliable statistics, we conducted additional experiments in which sperm samples were further diluted 1:10 and injected into the microfluidic chip in a controlled manner. This approach reduced the number of cells in the chamber to one or two, allowed us to observe the complete dynamics of individual cells, minimized cell-cell interactions, and reduced possible rheological heterogeneity arising at high cell densities. Under these conditions, we tracked 48 sperm (across 5 samples) after they entered the chamber. Rolling was suppressed in all of them upon entry, and they initially exhibited circular motion. However, 18 out of the 48 cells resumed rolling after a while, leading to a switch to wandering motion, followed by intermittent rolling

suppression at later times, resulting in repeated transitions between circling and wandering periods (Figs. 2a and S3, Movie S3, Tables S2–S6).

Such transitions were evident in the rapid changes in $\Omega$ from zero to nonzero values and vice versa (Figs. 2b and S2). The average times spent in the circling and wandering periods were $\tau_C = 67 \pm 23$ s and $\tau_W = 35 \pm 14$ s ($N = 18$ across 5 samples, Fig. S2b), respectively, and $\Omega$ remained constant across different constituent circular periods (Figs. 2b and S4). Consistent with the statistical picture obtained for pure circling and pure wandering, $D_{eff}$ inferred for the constituent circling periods was significantly smaller than that of the constituent wandering periods (Fig. S5). Further, the sperm persistence length during wandering periods—the average distance over which the cell maintains its direction before reorienting—was significantly greater ($l_p = (0.95 \pm 0.50) \times 10^3$ μm, $N = 34$) than that measured during pure wandering in TALP under the same 6.0 mM caffeine treatment ($l_p = (0.08 \pm 0.04) \times 10^3$ μm, $N = 50$). This persistence was comparable to that observed during pure wandering in TALP with milder caffeine stimulation (3.0 mM, $l_p = (0.55 \pm 0.40) \times 10^3$ μm, $N = 50$), yet it

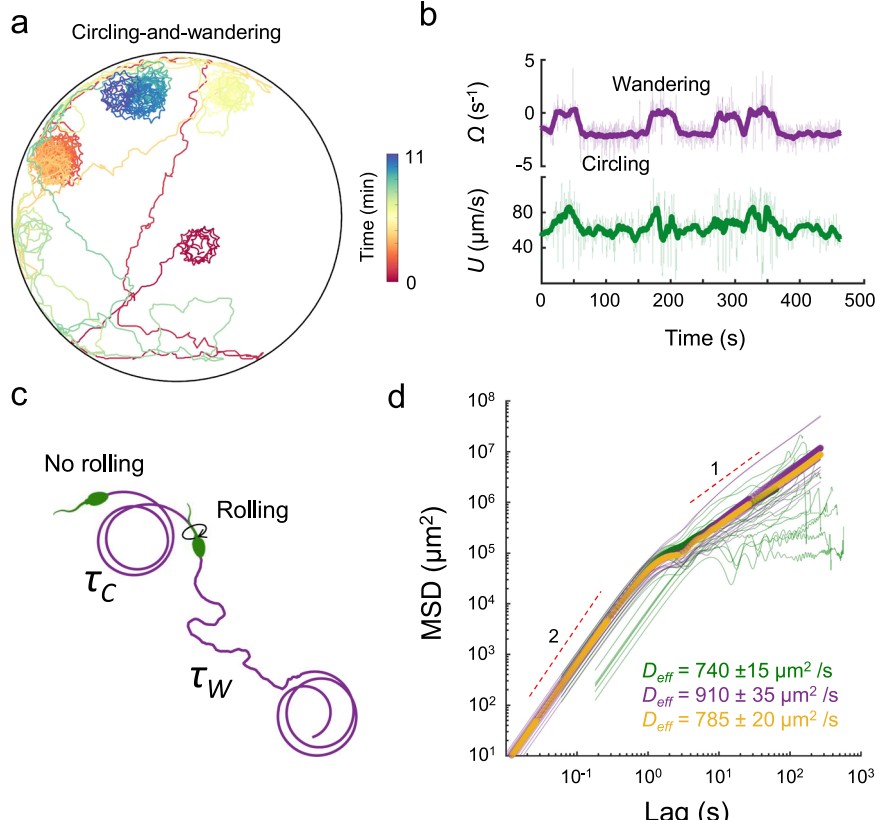

**Fig. 2 | Sperm circling-and-wandering. a** A representative trajectory of a circling-and-wandering sperm. The chamber radius is $R = 600\,\mu m$. **b** Instantaneous translational $U$ and rotational $\Omega$ speeds corresponding to the trajectory in (**a**). Real-time measurements are shown as thin lines, while the smoothed values are shown as thick lines. **c** Sketch of the model system, where $\tau_C$ and $\tau_W$ denote the average time sperm spend in each phase, respectively. **d** Experimental measurements and theoretical predictions for the MSD vs. lag time. The green line corresponds to the average experimental MSD, and the purple line corresponds to the average theoretical MSD. The transparent lines represent the experimental measurements and theoretical estimates for individual trajectories. Motility parameters measured from the velocity correlation function of individual trajectories have been used as input for the theory in the purple curve. The yellow curve corresponds to the theoretical MSD obtained from the motility parameters averaged over all individual trajectories. Source data for **b** and **d** are provided as a Source Data file.

remained markedly lower than the persistence length of progressive motion in TALP ($l_p = (2.10 \pm 0.65) \times 10^3\,\mu m$, $N = 50$).

To identify the physical basis of the intermittent circling-and-wandering dynamics, we measured the tilt of the sperm head during the different phases of motion and compared them to purely wandering or circling sperm. First, our measurements indicated that the head tilt in pure wandering was $\phi_W = 13 \pm 2°$, whereas during the wandering periods of circling-and-wandering this angle was $\phi_{CW} = 12 \pm 2°$ ($N = 5$), a difference that is not statistically significant ($p \approx 0.41$, Fig. S6a). Second, we examined whether the tilt angle before entering TALP + 1% PAM predicts whether a cell will exhibit pure circling or circling-and-wandering in the complex fluid. The initial tilt angle for cells that later exhibited pure circling was $\phi_C^{initial} = 12 \pm 4°$, while for cells that later exhibited circling-and-wandering it was $\phi_{CW}^{initial} = 12 \pm 2°$, a difference that is not statistically significant ($p \approx 0.83$, Fig. S6b) These observations let us conclude that the tilt angle remained approximately constant across all wandering motions.

To test whether transitions between circling and wandering arise from spatial heterogeneities in TALP+1% PAM, including possible transitions between Newtonian and non-Newtonian regions in the fluid, we added fluorescent microspheres (1 μm diameter) to the medium and tracked their motion for 5 min at 10 FPS across multiple chamber locations (Fig. S7a). From eight regions and more than one hundred microspheres per region, the mean

MSDs collapsed and showed consistent long-timescale diffusive behavior (Fig. S7b). The resulting effective diffusivity was $D_{eff} = (7.5 \pm 0.5) \times 10^{-3}\,\mu m^2\,s^{-1}$, corresponding to an effective viscosity of $60 \pm 4\,mPa\,s$ (Fig. S7c). We should note that the error bar lies below the uncertainty in our measurements of both diffusivity and viscosity within each region. This viscosity range is orders of magnitude higher than that of TALP($\approx 1\,mPa\,s$), where pure wandering occurs. These findings indicate that the emergence of circling-and-wandering reflects an intrinsic feature of hyperactivated motion in a moderately heterogeneous complex fluid, rather than an artifact of sharply separated Newtonian and non-Newtonian domains. Overall, our results suggest that a more complex mechano-chemical effect is the cause of the transitions between circling and wandering in complex fluids.

**Modeling circling-and-wandering.** To further characterize the experimentally-observed circling-and-wandering motility, we developed a stochastic model for the coarse-grained dynamics. We used a renewal theory[56,57] (see "Methods"), which provided a prediction for the MSD and the long-time effective diffusivity:

$$D_{eff}^{CW} = D + \frac{U^2\left(\tau_C^2\left(\tau_W\left(\tau_W\left(D_{rot}^2 + \Omega^2\right) + \lambda\right) + 1\right) + \tau_C^3 D_{rot}(\lambda\tau_W + 1) + \tau_W^2(2D_{rot}\tau_C + 1)\right)}{2(\lambda\tau_W + 1)(\tau_C + \tau_W)\left(\tau_C^2\left(D_{rot}^2 + \Omega^2\right) + 2D_{rot}\tau_C + 1\right)},$$

(1)

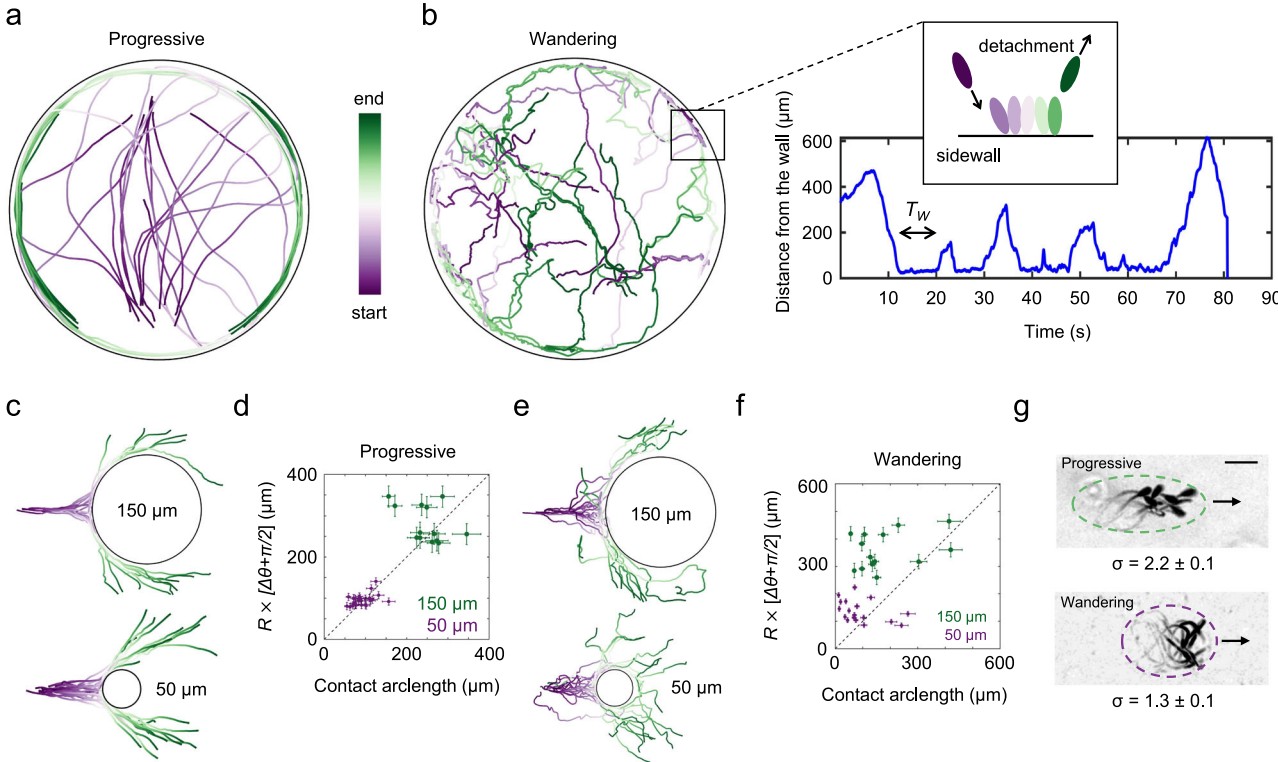

**Fig. 3 | Sperm-wall interactions in the progressive and wandering phases.**
**a** Progressive motility in TALP allows for entrapment around the circular sidewall. The chamber radius is $R = 600\,\mu m$. **b** The wandering phase allows sperm to explore the area enclosed by the circular sidewall. After each collision with the sidewall, sperm detach after a transitory dwell time $T_W$. **c** The swimming direction is deterministically rectified by pillars with $R = 150\,\mu m$ and $R = 50\,\mu m$. **d** Sperm contact arc length on the pillar is linearly correlated with the deflection angle $\Delta\theta$. The error bars reflect measurement uncertainty, not the standard deviation of replicates. **e** Wandering sperm scatter from the pillars, with $R = 150\,\mu m$ and $R = 50\,\mu m$, in a stochastic manner. **f** Sperm contact arclength on the pillar is not correlated with the deflection angle $\Delta\theta$. The error bars reflect measurement uncertainty, not the standard deviation of replicates. **g** Sperm aspect ratio $\sigma$ in progressive and wandering phases. Source data for **d** and **f** are provided as a Source Data file.

where $U$ denotes the swim speed and $D$ is the Brownian translational diffusivity. Further, we introduced $\lambda = D_{rot} + \tau^{-1}$, which depends on the rotational diffusivity $D_{rot}$ and persistence time $\tau$ of the wandering phase. We note that these parameters can be independently measured from experiments. The expression for the effective diffusivity [Eq. (1)] recovers coefficients for the pure individual phases ($D_{eff}^W$ and $D_{eff}^C$) by setting $\tau_C = 0$ ($\tau_W = 0$) and taking the limit of $\tau_W \to \infty$ ($\tau_C \to \infty$), respectively (Fig. S9b). The MSD (with independently measured motility parameters as inputs) agrees with our experimentally measured MSD, averaged over all trajectories, thus confirming that the coarse-grained theory describes the observed trajectories and serves as a model for future investigations (Fig. 2c, d). A comparison of $D_{eff}$ across all phases of motility shows that while pure wandering has higher effective diffusivities than pure circling, circling-and-wandering lies in between, suggesting its potential to bridge large-scale exploration and local exploitation strategies.

## Sperm-wall interactions: from concave to convex boundaries

To study sperm migration in complex geometries, we first quantified the swimming dynamics near concave boundaries (sidewall of the chamber) and convex (circular) pillars with different radii ($R = 150\,\mu m$ and $R = 50\,\mu m$) under the environmental conditions discussed earlier.

**Sliding of progressive sperm along boundaries.** Near concave boundaries, progressive sperm in TALP aligned with the boundary upon collision and swam along it until the end of the measurement, consistent with previous observations[4] (Fig. 3a). A similar pattern was observed near the circular pillars. In particular, sperm swimming directions were rectified upon collision with pillars; consequently, they swam along the pillar for only a short arclength, which was shorter for smaller pillars ($N = 39$, Fig. 3c). By measuring the cumulative changes in the direction of sperm swimming $\theta_{cumulative}$, i.e., the overall deflection angle $\Delta\theta$ (Fig. S8a), we found that pillars rectified the swimming direction in a deterministic manner, as the deflection angle and contact arclength were strongly correlated (Fig. 3d).

**Scattering of wandering sperm from boundaries.** The interaction of wandering sperm (TALP with 6.0 mM caffeine) with concave boundaries was significantly different from that of their progressive counterparts. They did not align with the sidewall but remained near the contact point until they were scattered away (Movie S4), swam through the chamber interior, and collided with the boundary at another point (Fig. 3b). We quantified this observation by measuring the distance between the sperm and the sidewall, as well as the distribution of dwell times, which had an average of $T_W = 13.8 \pm 14.8$ s ($N = 59$, Fig. S8e).

Similarly, wandering sperm remained near the contact point when they collided with the convex pillars until they scattered away (Figs. 3e and S8b, c and Movie S5). Unlike progressive sperm, the contact arclength on the pillar was not correlated with the deflection angle (Fig. 3f), and the relative standard deviation of the deflection

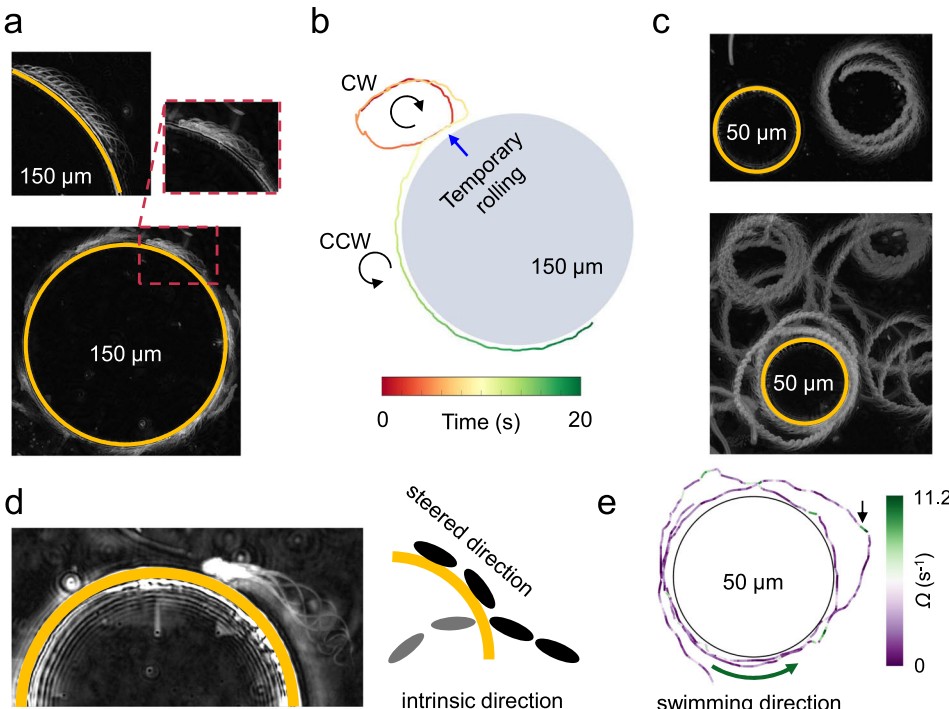

**Fig. 4 | Sperm-wall interactions in the circling phase. a** Without hyperactivation in 1%PAM, sperm in the circling phase feature large radii and become trapped around pillars with $R = 150$ μm. Entrapment of the first sperm (top) occurs within 1 min of entry into the chamber, increasing to 30 sperm within 3 min. **b** Temporary rolling events can change the chirality of circling motility in 1%PAM, allowing transitions from scattering to entrapment or vice versa. CW and CCW denote clockwise and counterclockwise chirality, respectively. **c** With hyperactivation (6.0 mM caffeine + 1%PAM), circling motility features tight circles, and sperm become trapped around $R = 50$ μm pillars. **d** When trapped, the sperm's intrinsic swimming direction is not necessarily aligned with the steered swimming direction, reducing its rotational speed. **e** A representative trajectory of a sperm circling around a pillar, color-coded with respect to its rotational speed. The arrow indicates a high rotational speed (deep bending in the flagellum) when the sperm is not in contact with the pillar.

angle was higher than that of progressive sperm ($N = 73$, Fig. S8d), highlighting the 'noisy' scattering of wandering sperm from the pillars.

In general, the random nature of wandering motility generates substantial orientational fluctuations that prevent sperm from aligning with boundaries. Furthermore, previous models suggested that the aspect ratio of a microswimmer determines its alignment dynamics when it collides with a boundary[58,59]. We then measured the aspect ratio, $\sigma = W/L$ (with width $W$ and length $L$), of progressive ($\sigma = 2.3 \pm 0.2$) and wandering ($\sigma = 1.3 \pm 0.2$) sperm, which revealed that the effective shape of the microswimmer changed from an ellipse in the progressive phase to a more spherical shape in the wandering phase (Figs. 3g and S9a, b). Therefore, wandering sperm are expected to experience less aligning torque from the sidewall (Fig. S9c). We note that the lower aspect ratio during wandering results from changes in the shorter wavelength and higher amplitude of the beating pattern.

**Entrapment of circling sperm by convex boundaries.** We then studied the interaction of the circling phase with pillars in TALP + 1% PAM. Without caffeine treatment, sperm swam in large circles and interacted with $R = 150$ μm pillars. Typically, when the curvature of the swimming path was positive (with respect to the pillar) at the contact point, the sperm became trapped; otherwise, they scattered from the pillar. Local contacts led to the alignment of the swimming direction along the pillar's periphery, resulting in persistent entrapment. The first entrapment event occurred within 0.5–1 min of sperm entry into the chamber, with up to

~30 and ~100 cells becoming trapped within 3 and 10 min, respectively (Fig. 4a and Movie S6). The trapped sperm exhibited non-uniform chirality, with 85 ± 11% displaying counterclockwise (CCW) motion (Fig. S10a). We also noticed that temporary rolling events at the contact point reversed the swimming chirality, and consequently transitioned sperm-wall interactions from scattering to entrapment, highlighting that having positive curvature at the contact point is key for entrapment (Figs. 4b and S10b, Movie S7).

However, a positive curvature at the contact point alone was insufficient for entrapment. Sperm with high-radius circular paths scattered from $R = 150$ μm pillars after partial rectification (Fig. S11a). Our measurements suggest that the maximum trajectory radius for entrapment was 520 ± 57 μm, and the minimum was 92 ± 11 μm (Fig. S11b). Without caffeine treatment, the circling sperm did not show a significant interaction with the $R = 50$ μm pillars within the 10-min observation period, indicating that the ratio between the radius of the pillar and the radius of circling is crucial for entrapment. To quantify these observations, we measured the radii of circular trajectories without caffeine (Fig. S11c) and found that the average radius (1.2 ± 0.7 mm) exceeded significantly $R = 50$ μm, explaining the lack of interaction with smaller pillars. However, a subset of cells had trajectories within the trapping range (>92 μm and <520 μm) for $R = 150$ μm pillars, accounting for approximately 11.3% of the cells, the estimated fraction of trapped sperm.

In TALP + 1%PAM + 6.0mM caffeine swimming medium, circling sperm did not become trapped around $R = 150$ μm pillars. This is not surprising, as after measuring the radii of the

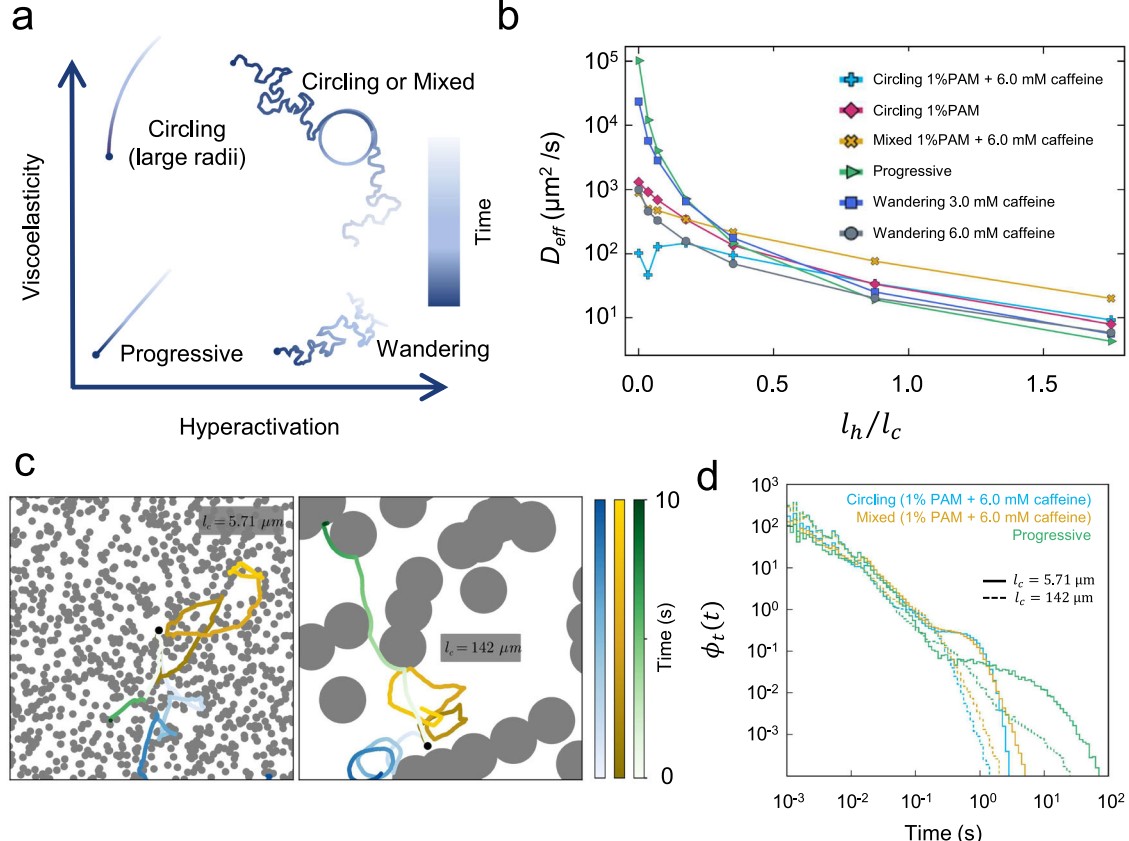

**Fig. 5 | Sperm motion in porous media. a** Different swimming behaviors promoted by hyperactivation and the viscoelasticity of the swimming medium. **b** Effective diffusion coefficient ($D_{eff}$) measured for sperm moving in porous media under different conditions. Here, $l_h$ denotes the sperm head length and $l_c$ represents the average chord length. **c** Typical trajectories of progressive (green), circling (blue), and mixed-phase (yellow) motion in two different porous media. The obstacle radii are 2 μm and 50 μm in the left and right panels, respectively. Color coding corresponds to elapsed time. **d** Trapping time distributions, $\phi_t(t)$, for the three phases in the same environments as in (**c**).

circling trajectories at the population level (Fig. S11d), we found that the average radius (36 ± 10 μm) was below the minimum entrapment radius (92 ± 11 μm) for the pillars $R = 150$ μm. In contrast, and as expected, entrapment occurred around the $R = 50$ μm pillars in 10 min (1–3 cells per pillar, Fig. 4c, top at $t = 0$ and bottom at $t \approx 10$ min, Movie S6). To further quantify this observation, we measured the velocity correlation function for $N = 7$ sperm before and after entrapment to determine their rotational speed. We found that $\Omega$ decreased significantly upon entrapment ($\Omega_{trapped} = 1.4 \pm 0.1\,s^{-1}$ vs. $\Omega_{free} = 2.0 \pm 0.5\,s^{-1}$, $p$ value = 0.01; Fig. S12), likely due to misalignment between the intrinsic swimming direction and the boundary constraint (Fig. 4d, e).

**Sperm transport in a geometrically complex environment**
Having established an experimental classification of hyperactivated motility in both Newtonian and non-Newtonian fluids, we investigated the impact of the observed motility patterns, ranging from progressive swimming to circling to circling-and-wandering dynamics, on the spreading of active agents in a geometrically complex environment. Therefore, we model the different types of motion as stochastic processes, thereby coarse-graining the physical mechanisms that give rise to such dynamics. Different phases of sperm motility (Fig. 5a) are described using the minimal models we proposed earlier, with experimentally measured parameters as input (see "Methods", Table 1). The geometrically complex environment is modeled as a

two-dimensional porous medium composed of overlapping discs of radius $R_0$, reflecting the quasi-2D, crowded nature of the folds of the FRT. We keep a constant packing fraction $\eta = N\pi R_0^2/L^2$ (with number of obstacles $N$ and system size $L$) and vary the disc radius $R_0$[60]. This generates porous media with varying average chord lengths, $l_c = \pi R_0/(2\eta)$, and thus different numbers of corners and dead-end passages (Fig. 5c).

We quantified sperm dynamics by measuring the MSD, which grows linearly with time at long times, $\langle|\Delta\vec{r}(t)|^2\rangle = 4D_{eff}t$, allowing us to extract $D_{eff}$ as a measure of the dispersion of cells in the porous medium (Fig. S13). In dilute media (corresponding to large average chord lengths $l_c$), our simulations show that $D_{eff}$ is largest for progressive and smallest for circling sperm (1% PAM + 6.0 mM caffeine), while the mixed phase lies in between, confirming our previous statistical characterization of motility in free space (Fig. 5b). Decreasing the average chord length $l_c$ leads to a stark decrease in $D_{eff}$ for the progressive phase by up to four orders of magnitude, making the progressive phase the least effective in a porous medium. The effective diffusivity $D_{eff}$ also decreases for the other phases, but the effect is not as pronounced. Interestingly, our results further show that the mixed phase is only weakly affected by changes in the environment and becomes the most efficient transport strategy as the average chord length is decreased.

This behavior can be rationalized by inspecting the trajectories in a medium with a long average chord length, $l_c = 142$ μm (Fig. 5c). Here, progressive sperm are able to explore large areas through their

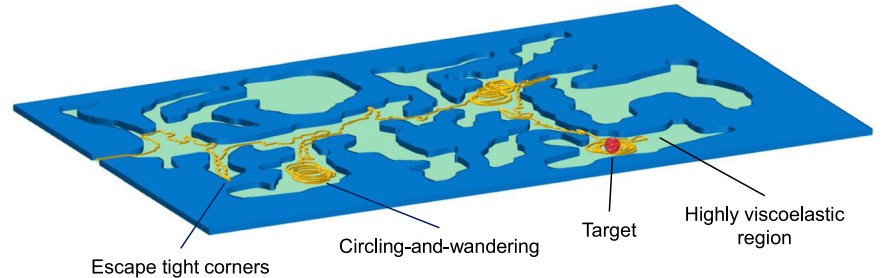

**Fig. 6 | Sperm circling-and-wandering in the FRT.** Schematic of how circling-and-wandering swimming behavior may facilitate sperm migration within the FRT.

almost straight swimming motion, while circling sperm circle in the open pore space without migrating far from the initial position. During the circling-and-wandering phase, sperm can explore the environment more than those with pure circling, but their continuous random orientational changes lead to an overall smaller displacement than that of the progressive phase. This behavior changes drastically when the average chord length is decreased to $l_c = 5.71\,\mu m$ (which is comparable to the sperm head size), where progressive sperm remain stuck in the corners of the porous matrix, while the other motility patterns lead to larger exploration of the pore space. We emphasize that the mixed phase remains more efficient than the pure circling, as the wandering periods help the cell move farther.

These observations can also be quantified by measuring the trapping times, i.e., the time sperm are trapped at obstacle boundaries and in narrow channels where they barely move (Fig. 5d). The distributions show distinct features for the different phases and indicate long trapping phases for progressive sperm. For the mixed and circling phases, the trapping times are shorter and the decay of the distribution is faster. This is due to cells being able to reorient, which allows them to escape from tight corners and boundaries, thus reducing the trapping times[61,62]. From our simulation results, we may conclude that hyperactivated motility, regardless of fluid properties, shows enhanced dispersion compared to progressive motility; that is, randomness in swimming behavior is crucial for spreading in confined and complex geometries, whereas progressive motility is more suitable for regions with less complexity.

## Discussion

In summary, our results show that hyperactivation enables sperm to adjust their motility in vitro, which yields distinct patterns that depend on the viscoelasticity of the surrounding fluid. In a Newtonian fluid, hyperactivated sperm display wandering behavior, while at high viscoelasticity, they shift to circling or circling-and-wandering modes. In our experiments, wandering sperm, being less affected by geometric constraints, facilitated broader exploration of the landscape. In contrast, circling sperm, characterized by lower effective diffusivity, tended to remain near obstacles. Moreover, our simulations showed that as geometric complexity increases, circling-and-wandering achieves higher effective diffusivity than wandering or circling alone. Although the different swimming behaviors were observed in microfluidic chambers and under conditions that generically mimic the tract's environment, they may hint at how sperm could migrate through viscoelastic regions in more complex biological settings. For instance, the circling-and-wandering mode, in particular, reflects a balance between exploring large areas and localized exploitation (Fig. 6). This balance might enhance search efficiency, resembling

intermittent search strategies observed across diverse biological systems[63,64].

It is important to note that our theoretical predictions for the mixed phase rely on the choice of exponential distributions for the circling and wandering times, which nicely described our data for the MSDs without any fitting parameters. Our theory can, however, be readily applied to arbitrary distributions that have a finite mean. Thus, future work may focus on deriving higher-order moments for the displacements or the intermediate scattering functions (i.e., the Fourier transform of the probability densities)[56,65], providing access to the full spatiotemporal dynamics. The latter may enable insights into the role of the distributions for sperm dynamics. Another assumption of our theory is that sperm randomly change their swimming direction at the transition between two phases. This is reasonable because both phases—circling and wandering—already lead to substantial changes of the agent's swimming direction at smaller time scales ($1/\Omega \approx 0.5\,s \lesssim \tau_C \approx 70\,s$ and $\tau \approx 0.7\,s \lesssim \tau_W \approx 40\,s$) and hence one additional change of swimming direction at the transition does not contribute to the results (Fig. S14).

The circling dynamics of sperm, quantified in this work, are inherent to a large class of active agents near boundaries, including bacteria[66], where the rotation of the flagella bundle and the counter-rotation of the cell body lead to circular motion as a result of hydrodynamic interactions with the wall, and self-propelled colloids[67,68], whose shape-asymmetries introduce circular dynamics. Despite geometrical effects, circular motion has also been identified in spherical, achiral Janus colloids immersed in a non-Newtonian medium[69], whose angular velocity can switch sign. In our work—in addition to geometry and rheology—the angular velocity strongly depends on the amount of caffeine and, consequently, a more complex interplay of biophysical and mechanical features appears to be important that needs to be unraveled in the future. Due to these intricate dynamics, sperm interactions with discs differ from previous experiments of self-propelled rods[70] and theoretical work on microswimmers[71], where the interplay of noise and hydrodynamic interactions governs the detention times at the boundaries. In particular, how steric interactions of the long sperm flagellum, together with these hydro-chemical effects, result in trapping by obstacles, represents an interesting research question.

We speculate that circling-and-wandering represents a candidate framework for understanding mammalian sperm chemotaxis. In analogy to the well-studied bacterial running-and-tumbling model[56,72], one could view circling-and-wandering as stochastic transitions between local reorientation and long-range displacement. However, we should stress that despite similarities between bacterial tumbling and sperm circling, a key difference exists: whereas bacterial tumbling occurs in the order of milliseconds (or a fraction of the run time), sperm circling

unfolds over minutes, with a duration comparable to that of the wandering phase. Therefore, and as we suggested earlier, the circling phase may function as a local exploitation strategy rather than a mere change of the swimming direction.

In the context of chemotaxis, the next questions concern what drives circling-and-wandering behavior and how environmental factors can modulate it, for example, by tuning the transition frequency between circling and wandering periods. This requires identifying potential coupling mechanisms between chemical effects due to hyperactivation and mechanical effects stemming from the fluidic surroundings. At the same time, progress along this direction demands stronger statistical power in characterizing circling-and-wandering motility. Although we report this behavior, our current approach yielded 18 out of 48 hyperactivated tracks despite extensive experimental effort. Therefore, future progress will rely on the development of new experimental methods that can visualize and quantify the motion of sufficiently large sperm populations in complex fluids for minutes to hours[73].

Although we relied on a chemical agonist to induce hyperactivation, recent evidence indicates that activation of CatSper, and thus hyperactivation, can also arise from thermal inputs[74]. This suggests that hyperactivation may be a general feature of both chemotactic and thermotactic responses, raising the possibility—yet to be tested—that circling-and-wandering contributes to thermotaxis as well. Likewise, directional cues such as chemical gradients, fluid flow within the oviduct, or spatial variations in viscoelasticity may influence circling-and-wandering motility; however, confirming this will require future experiments conducted under physiologically representative conditions, particularly within the mucus of the female reproductive tract.

Finally, although our findings arise from controlled microfluidic environments, they may inform the design of assisted reproductive technologies, especially those that leverage or select for hyperactivated motility to improve the efficacy of current approaches[75–77].

## Methods

### Microfluidic device

Our microfluidic device comprised a straight microchannel (width ~400 μm) connected to circular Hele-Shaw chambers on either side, each with a diameter of 1200 μm. For wall-interaction experiments, we fabricated concentric pillars with diameters of 100 and 300 μm within the chambers. The height of the device was 30 μm throughout. The chambers were initially filled with the swimming media, e.g., TALP + caffeine or TALP + 1% PAM + caffeine. Later, the sperm sample was injected into the main channel using gravity, and by reducing the flow, sperm cells were able to swim inside the chambers without perturbations caused by external fluid flow. Microfabrication was carried out using standard soft lithography techniques.

### Sperm preparation and chemicals

Commercially available cryopreserved bull sperm samples (5.5–6.5 years of age) diluted in egg yolk extender were generously donated by URUS Holding Company, Ithaca, NY. The ejaculate concentration was $50 \times 10^6$ sperm per straw (0.25 mL) with a pre-freeze motility of 65%. A combination of gentamicin, tylosin, lincomycin, and spectinomycin was added to the semen as antibiotics prior to cryopreservation. For each experiment, two straws were thawed in a 38 °C water bath for 30 s, diluted with 1 mL TALP, incubated at 38 °C and 5% $CO_2$ (Thermo Fisher) for 30 min, and then used immediately. The sperm samples and microfluidic device were maintained at 38 °C throughout the experiments using a heated glass microscope plate (Bioscience Tools). The following recipe was used for making TALP:

NaCl (110 mM), KCl (2.68 mM), $NaH_2PO_4$ (0.36 mM), $NaHCO_3$ (25 mM), $MgCl_2$ (0.49 mM), $CaCl_2$ (2.4 mM), Hepes buffer (25 mM), glucose (5.56 mM), pyruvic acid (1.0 mM), penicillin G (0.006% or 3 mg/500 mL), and bovine serum albumin (20 mg/mL). Long-chain PAM with a molecular weight of 5–6 MDa was added to make the standard medium viscoelastic. To prevent contamination, sperm preparation and loading into the microfluidic devices were carried out in a biosafety cabinet, and all buffers used in this experiment were kept sterile.

### Microscopy

The hyperactivated motility of sperm inside the circular chambers was observed and recorded using a Zeiss phase-contrast microscope (10× objective) equipped with a Neo sCMOS high-speed digital camera (Andor), with image acquisition performed using NIS Elements software (Version 4.0; Nikon). Under phase optics, the rolling of bull sperm could be detected as variations in the light intensity of the paddle-shaped head. To minimize the effect of ambient temperature on sperm motion, the microfluidic devices were maintained at 38 °C using a heated glass slide (Bioscience Tools). Flash red polystyrene beads with an average diameter of 1 μm (Bangs Laboratories) were used for microrheological measurements in TALP+1% PAM. For fluorescence imaging of the beads, a Nikon Eclipse Ti microscope (10× objective) with a sCMOS high-speed digital camera (Hamamatsu) recorded bead positions at 10 FPS.

### Image processing

The acquired videos were processed to extract sperm trajectories using TrackMate and a custom MATLAB code. The trajectories obtained manually and analyzed in Matlab needed to be rescaled after acquisition. Each trajectory was then coarse-grained by averaging every five consecutive points to remove rapid oscillatory motion in the sperm head. Using the smoothed trajectories, we then calculated the translational velocity, $\vec{V}(t)$, and the normalized velocity autocorrelation function:

$$C_{vv}(t, \alpha) = \frac{\langle \vec{V}(t', \alpha) \cdot \vec{V}(t' + t, \alpha) \rangle}{\langle \vec{V}(t', \alpha) \cdot \vec{V}(t', \alpha) \rangle} \qquad (2)$$

where $\alpha = 5$ was used as a coarse-graining parameter. This velocity correlation function was then used to estimate values for rotational speed and relaxation times. We also measured the sperm reorientation angle,

$$\theta(t) = \tan^{-1}\left(\frac{\vec{e}(t) \times \vec{e}(t + \delta t)}{\vec{e}(t) \cdot \vec{e}(t + \delta t)}\right), \qquad (3)$$

with the cumulative angle given by $\theta_{\text{cumulative}}(t) = \sum_0^t \theta(t)$.

After demonstrating that wandering, circling, and mixed phases are all diffusive in nature, we inferred diffusion coefficients based on the MSD, or on rotational and translational speeds and relaxation times for trajectories shorter than 150 points.

### Theory and simulation

**Theoretical derivation of the effective diffusivities for the mixed phase: a renewal approach.** Here, we first discuss the MSDs of the two different phases: $\langle |\Delta\vec{r}(t)|^2 \rangle_{C,W} = \langle |\vec{r}(t) - \vec{r}(0)|^2 \rangle$ with initial position $\vec{r}(0)$ and noise average $\langle \cdot \rangle$. We note that the MSDs for the circling[54,55] and wandering phases[78] have been previously reported and we refer the reader to previous work for a derivation. The MSD for the circling phase, where the sperm is modeled as an active Brownian circle

simmer, evaluates to[54,55]

$$\langle|\Delta\vec{r}(t)|^2\rangle_C = 4Dt + \frac{2U^2}{\left(D_{rot}^2+\Omega^2\right)^2}\exp(-D_{rot}t)\Big[\exp(D_{rot}t)\left(D_{rot}^2(D_{rot}t-1)+\Omega^2(D_{rot}t+1)\right)$$
$$+\left(D_{rot}^2-\Omega^2\right)\cos(\Omega t)-2D_{rot}\Omega\sin(\Omega t)\Big],$$

(4)

with swim speed $U$, angular velocity $\Omega$, and rotational and translational diffusivities, $D_{rot}$ and $D$, respectively. The wandering phase is modeled as a persistent random walk, where a particle randomly changes its swimming direction at exponentially distributed times, $\exp(-t/\tau)/\tau$ with average time between two events $\tau$[78]. Thus, the MSD reads

$$\langle|\Delta\vec{r}(t)|^2\rangle_W = 4Dt + \frac{2U^2}{D_{rot}+\tau^{-1}}\left[\exp\left(-t(D_{rot}+\tau^{-1})\right)+(D_{rot}+\tau^{-1})t-1\right],$$

(5)

where the effects of translational and rotational diffusion are also included. We note that the contributions of rotational diffusion and the random change of swimming direction are additive at the level of the MSD and cannot be distinguished from one another. These two MSDs serve as input to compute the MSD in the mixed phase. The latter has not been derived before, and, therefore, we present its derivation using a renewal framework.

The mixed phase consists of alternations of circling ($C$) and wandering ($W$) phases. We assume the agent spends an average time $\tau_C$ and $\tau_W$ in each of both phases, respectively, and that at the start of a new phase, it moves along a new, random direction. A theoretical framework, allowing to couple two distinct phases, has been previously reported in terms of renewal equations[79] and, among others, applied to study the run-and-tumble motion of bacteria in free space[56,57,80] and their hop-and-trap dynamics in porous media[62]. Here, we denote by $P(\vec{r},t)$ the probability density to be in a mixed phase at position $\vec{r}$ at time $t$. It can be decomposed as a sum of the probability to be in a circling phase $P_C(\vec{r},t)$ and a wandering phase $P_W(\vec{r},t)$: $P(\vec{r},t) = P_C(\vec{r},t)+P_W(\vec{r},t)$. To characterize the circling phase, we require two ingredients: the probability to move a distance $\vec{r}$ during time $t$ in the chiral mode, $\mathbb{P}_C(\vec{r},t)$, and the probability that the circling phase ends at time $t$, $\varphi_C(t) = \exp(-t/\tau_C)/\tau_C$. Similarly, for the wandering phase, we introduce: the probability to move a distance $\vec{r}$ during time $t$ in the persistent-random-walk mode, $\mathbb{P}_W(\vec{r},t)$, and the probability that the wandering phase ends at time $t$, $\varphi_W(t) = \exp(-t/\tau_W)/\tau_W$. Assuming that the agents are in a non-equilibrium stationary state at time $t$, we need to account for the probability that the cells have spent their entire time up to $t$ in $C$ ($W$), respectively:

$$P_{C,W}^0(\vec{r},t) = p_{C,W}\mathbb{P}_{C,W}(\vec{r},t)\int_t^\infty d\tau\varphi_{C,W}^0(\tau)/\tau_{C,W},$$

(6)

where $p_C = \tau_C/(\tau_C+\tau_W)$ and $p_W = 1-p_C$ denote the fraction of time spent in $C$ and $W$, respectively. Further, $\varphi_{C,W}^0 := \int_t^\infty d\tau\,\varphi_{C,W}(\tau)$ denotes the survival probability, i.e., the probability that the agent remains in $C$ ($W$). The time integral in Eq. (6) corresponds to the probability density to switch from one to the other phase for the first time. See ref. 57 for more detailed explanations.

The two phases, $P_C(\vec{r},t)$ and $P_W(\vec{r},t)$, are then described by the integral equations:

$$P_C(\vec{r},t) = P_C^0(\vec{r},t) + \int_0^t d\tau\int_{\mathbb{R}^2}d\vec{\ell}\,C(\vec{r}-\vec{\ell},t-\tau)\varphi_C^0(\tau)\mathbb{P}_C(\vec{\ell},\tau),$$

(7a)

$$P_W(\vec{r},t) = P_W^0(\vec{r},t) + \int_0^t d\tau\int_{\mathbb{R}^2}d\vec{\ell}\,W(\vec{r}-\vec{\ell},t-\tau)\varphi_W^0(\tau)\mathbb{P}_W(\vec{\ell},\tau),$$

(7b)

where we denote by $C(\vec{r},t)$ and $W(\vec{r},t)$ the probabilities (per unit time) to start a circling or wandering phase, respectively. Thus, the probability to be in $C$ ($W$) at time $t$ and position $\vec{r}$ is the sum of the probability to be in $C$ ($W$) without ever having been in $W$ ($C$) before and the sum over all previous changes between different phases, encoded in the integral. We note that neglecting $P_{C,W}^0(\vec{r},t)$ corresponds to a model where the cells start the $C$ ($W$) phase at time $t$, which is rather hard to observe experimentally.

Next, we introduce equations of motion for the unknowns $C(\vec{r},t)$ and $W(\vec{r},t)$, which couple the two processes:

$$C(\vec{r},t) = C^1(\vec{r},t) + \int_0^t d\tau\int_{\mathbb{R}^2}d\vec{\ell}\,W(\vec{r}-\vec{\ell},t-\tau)\varphi_W(\tau)\mathbb{P}_W(\vec{\ell},\tau)$$

(8a)

$$W(\vec{r},t) = W^1(\vec{r},t) + \int_0^t d\tau\int_{\mathbb{R}^2}d\vec{\ell}\,C(\vec{r}-\vec{\ell},t-\tau)\varphi_C(\tau)\mathbb{P}_C(\vec{\ell},\tau).$$

(8b)

Here, we have again accounted for the probability to start a $C$ ($W$) phase for the first time separately using

$$C^1(\vec{r},t) = p_W\mathbb{P}_W(\vec{r},t)\varphi_W^0(t)/\tau_W,$$

(9a)

$$W^1(\vec{r},t) = p_C\mathbb{P}_C(\vec{r},t)\varphi_C^0(t)/\tau_C.$$

(9b)

Moving to Fourier-Laplace space ($\vec{r}\to\vec{k}$ and $t\to s$) and using the convolution theorem, the set of equations has a formal solution[80]:

$$P_C(k,s) = P_C^0(k,s) + \mathcal{L}[\varphi_C^0(t)\mathbb{P}_C(k,t)](s)\frac{C^1(k,s)+W^1(k,s)\mathcal{L}[\varphi_W(t)\mathbb{P}_W(k,t)](s)}{1-\mathcal{L}[\varphi_C(t)\mathbb{P}_C(k,t)](s)\mathcal{L}[\varphi_W(t)\mathbb{P}_W(k,t)](s)},$$

(10a)

$$P_W(k,s) = P_W^0(k,s) + \mathcal{L}[\varphi_W^0(t)\mathbb{P}_W(k,t)](s)\frac{W^1(k,s)+C^1(k,s)\mathcal{L}[\varphi_C(t)\mathbb{P}_C(k,t)](s)}{1-\mathcal{L}[\varphi_W(t)\mathbb{P}_W(k,t)](s)\mathcal{L}[\varphi_C(t)\mathbb{P}_C(k,t)](s)},$$

(10b)

where $\mathcal{L}[f(t)](s) := \int_0^\infty dt\exp(-st)f(t)$ denotes the Laplace transform of a function $f(t)$. Since the problem is translationally invariant, the quantities depend on the wavenumber $k = |\vec{k}|$ only. To obtain the MSD for the mixed phase, we are interested in the small-wavenumber expansion of the probability density, which, in Laplace space, assumes the form:

$$P(k,s) = \int_0^\infty dt\,e^{-st}P(k,t) = \int_0^\infty dt\,e^{-st}\left[1-\frac{k^2}{4}\langle(\Delta\vec{r}(t))^2\rangle\right]+\mathcal{O}(k^4)$$

(11a)

$$= s^{-1} - \frac{k^2}{4}\mathcal{L}[\langle|\Delta\vec{r}(t)|^2\rangle](s)+\mathcal{O}(k^4).$$

(11b)

Therefore, we can also expand the propagators up to $\mathcal{O}(k^4)$ via

$$\mathbb{P}_{C,W}(k,t) = 1 - \frac{k^2}{4}\langle|\Delta\vec{r}(t)|^2\rangle_{C,W}+\mathcal{O}(k^4),$$

(12)

Inserting these expansions into Eqs. (10a) and (10b) and performing the Laplace transforms, allows calculating an analytical expression for $P(k,s)$ to second order in $k$ and hence the MSD in Laplace space, which can be transformed back to time analytically (Eq. (13) below) using Mathematica:

$$
\begin{aligned}
\langle |\Delta \vec{r}(t)|^2 \rangle = h^{-1} \Bigg[ & 2Dt + e^{-\frac{t}{\tau_C}} \frac{\tau_C \tau_W v^2}{\alpha \rho \delta} \left( \tau_C \tau_W D_{\mathrm{rot}}^2 - \rho D_{\mathrm{rot}} + \Omega^2 \tau_C \tau_W \right) + e^{-t\left(\frac{1}{\tau_W}+\frac{1}{\tau_C}\right)} \frac{\tau_C(\tau_C-\tau_W)\tau_W v^2}{\alpha^2 \gamma \kappa} \\
& \times \left( \tau_W(D_{\mathrm{rot}}\tau_W - 1) + \tau_C\left(D_{\mathrm{rot}}^2 \tau_W^2 + \Omega^2 \tau_W^2 - D_{\mathrm{rot}}(\tau^{-1}\tau_W + 2)\tau_W + \beta\right) \right) \\
& + \frac{v^2}{\alpha^2 \beta \epsilon} \left( D_{\mathrm{rot}}^2 \tau_C^2 \tau_W^4 + \left(\left(\frac{5\tau_C^3}{\tau} + \left(\Omega^2 \tau_W^2 + 3\right)\tau_C^2 + \tau_W^2\right)\tau_W^2 + D_{\mathrm{rot}}\tau_C^3(\beta\tau_C^2 + 5\tau_W \tau_C + 3\tau_W^2)\right) \right) \\
& + \frac{v^2}{\alpha \beta \epsilon} t \left( \left(\frac{\tau_W D_{\mathrm{rot}}}{\tau} + D_{\mathrm{rot}}\right)\tau_C^3 + \left(D_{\mathrm{rot}}^2 \tau_W^2 + \Omega^2 \tau_W^2 + \beta\right)\tau_C^2 + 2D_{\mathrm{rot}}\tau_W^2 \tau_C + \tau_W^2 \right) \\
& + \frac{v^2}{\alpha \beta^2 \rho \gamma} e^{-t\left(\frac{1}{\tau}+\frac{1}{\tau_W}\right)} \tau_W \left(\frac{\tau_W^3}{\tau} + \left(\frac{\tau_W}{\tau} - 1\right)\left(\frac{\tau_W \tau_C}{\tau} + \tau_C\right)^2 + \tau_C\left(\tau_W - \frac{2\tau_W^3}{\tau^2}\right)\right)\tau \\
& + e^{-t\left(D_{\mathrm{rot}}+\frac{1}{\tau_C}\right)} v^2 \tau_C \Bigg( \Big( \tau_C^6 \tau_W^4 D_{\mathrm{rot}}^8 + 4\tau_C^5 \tau_W^3(\tau_W - \tau_C)D_{\mathrm{rot}}^7 + \tau_C^4 \tau_W^2(2(\Omega^2 \tau_W^2 + 3)\tau_C^2 - 13\tau_W \tau_C + 5\tau_W^2)D_{\mathrm{rot}}^6 \\
& - 4\tau_C^4 \tau_W((\Omega^2 \tau_W^2 + 1)\tau_C^2 - 2\tau_W(\Omega^2 \tau_W^2 + 2)\tau_C + 3\tau_W^2)D_{\mathrm{rot}}^5 + \tau_C^2\Big((2\Omega^2 \tau_W^2 + 1)\tau_C^4 - \tau_W(23\Omega^2 \tau_W^2 + 9)\tau_C^3 \\
& + \tau_W^2(17\Omega^2 \tau_W^2 + 11)\tau_C^2 + 3\tau_W^3 \tau_C - 5\tau_W^4\Big)D_{\mathrm{rot}}^4 + \tau_C\Big(4\Omega^4 \tau_W^3 \tau_C^5 + (4\Omega^4 \tau_W^4 + 20\Omega^2 \tau_W^2 + 2)\tau_C^4 - \tau_W(44\Omega^2 \tau_W^2 + 5)\tau_C^3 \\
& + 4\tau_W^2(6\Omega^2 \tau_W^2 - 1)\tau_C^2 + 11\tau_W^3 \tau_C - 4\tau_W^4\Big)D_{\mathrm{rot}}^3 + \Big(-2\Omega^4 \tau_W^2(\Omega^2 \tau_W^2 + 3)\tau_C^6 - \Omega^2 \tau_W(7\Omega^2 \tau_W^2 + 8)\tau_C^5 \\
& + (11\Omega^4 \tau_W^4 + 32\Omega^2 \tau_W^2 + 1)\tau_C^4 + (\tau_W - 42\Omega^2 \tau_W^3)\tau_C^3 + \tau_W^2(20\Omega^2 \tau_W^2 - 7)\tau_C^2 + 6\tau_W^3 \tau_C - \tau_W^4\Big)D_{\mathrm{rot}}^2 \\
& + \Big(4\Omega^6 \tau_W^3 \tau_C^6 + 4\Omega^4 \tau_W(\tau_C^3 + \tau_W \tau_C^2 - 4\tau_W^2 \tau_C + 2\tau_W^3)\tau_C^3 + \Omega^2(\tau_C - \tau_W)^2(2\tau_C^2 - 5\tau_W \tau_C + 8\tau_W^2)\tau_C \\
& + (\tau_C - \tau_W)^2 \tau_W\Big)D_{\mathrm{rot}} + \Omega^2\Big(-(\tau_W^2 \Omega^3 + \Omega)^2 \tau_C^6 + (3\tau_W^3 \Omega^4 + \tau_W \Omega^2)\tau_C^5 + (-\Omega^4 \tau_W^4 + 5\Omega^2 \tau_W^2 + 1)\tau_C^4 \\
& - \tau_W(5\Omega^2 \tau_W^2 + 3)\tau_C^3 + \tau_W^2(\Omega^2 \tau_W^2 + 5)\tau_C^2 - 4\tau_W^3 \tau_C + \tau_W^4\Big)\Big) \cos(\Omega t) - \Omega\Big(2\tau_C^6 \tau_W^4 D_{\mathrm{rot}}^7 \\
& + 2\tau_C^5 \tau_W^3(3\tau_W - 4\tau_C)D_{\mathrm{rot}}^6 + 2\tau_C^4 \tau_W^2(3(\Omega^2 \tau_W^2 + 2)\tau_C^2 - 8\tau_W \tau_C + \tau_W^2)D_{\mathrm{rot}}^5 - 2\tau_C^3 \tau_W\Big((8\Omega^2 \tau_W^2 + 4)\tau_C^2 \\
& - \tau_W(7\Omega^2 \tau_W^2 + 9)\tau_C^2 - 2\tau_W^2 \tau_C + 6\tau_W^3\Big)D_{\mathrm{rot}}^4 + 2\tau_C^2\Big((3\Omega^4 \tau_W^4 + 8\Omega^2 \tau_W^2 + 1)\tau_C^4 - \tau_W(16\Omega^2 \tau_W^2 + 5)\tau_C^3 \\
& + (6\Omega^2 \tau_W^4 - 3\tau_W^2)\tau_C^2 + 16\tau_W^3 \tau_C - 9\tau_W^4\Big)D_{\mathrm{rot}}^3 + \tau_C\Big(-8(\tau_W^3 \Omega^4 + \tau_W \Omega^2)\tau_C^5 + 2(5\Omega^4 \tau_W^4 + 12\Omega^2 \tau_W^2 + 1)\tau_C^4 \\
& + (\tau_W - 20\Omega^2 \tau_W^3)\tau_C^3 + (8\Omega^2 \tau_W^4 - 22\tau_W^2)\tau_C^2 + 29\tau_W^3 \tau_C - 10\tau_W^4\Big)D_{\mathrm{rot}}^2 + 2\Big(\Omega^6 \tau_W^4 \tau_C^6 + \Omega^4 \tau_W^2(2\tau_C^2 - 8\tau_W \tau_C + 5\tau_W^2)\tau_C^4 \\
& + \Omega^2(\tau_C^4 - 5\tau_W \tau_C^3 + 5\tau_W^2 \tau_C^2 - 4\tau_W^3 \tau_C + 3\tau_W^4)\tau_C^2 - \tau_W(-2\tau_C^3 + 6\tau_W \tau_C^2 - 5\tau_W^2 \tau_C + \tau_W^3)\Big)D_{\mathrm{rot}} + 2\Omega^6 \tau_C^5 \tau_W^4 + (\tau_C - \tau_W)^2 \tau_W \\
& + \Omega^2 \tau_C(\tau_C - \tau_W)^2(2\tau_C^2 + \tau_W \tau_C + 2\tau_W^2) + 2\Omega^4 \tau_C^3 \tau_W^2(3\tau_C^2 - 4\tau_W \tau_C + 2\tau_W^2)\Big) \sin(\Omega t)\Bigg) \\
& - \frac{e^{-\frac{t}{\tau_W}}\tau_C \tau_W\big((D_{\mathrm{rot}}^2 \tau_W^2 + \Omega^2 \tau_W^2 - D_{\mathrm{rot}}(\tau^{-1}\tau_W + 2)\tau_W + \beta)\tau_C^2 + \tau_W(2D_{\mathrm{rot}}\tau_W - \frac{\tau_W}{\tau} - 2)\tau_C + \tau_W^2\big)\tau v^2}{\alpha\big(\kappa\tau_C^2 + 2\tau_W(D_{\mathrm{rot}}\tau_W - 1)\tau_C + \tau_W^2\big)} \\
& - \frac{v^2}{\alpha^2 \beta^2 \epsilon^2}\left(\left(\left(\frac{\tau_W D_{\mathrm{rot}}}{\tau} + D_{\mathrm{rot}}\right)\tau_C^3 + \left(D_{\mathrm{rot}}^2 \tau_W^2 + \Omega^2 \tau_W^2 + \beta\right)\tau_C^2 + \tau_W^2\right)\right. \\
& \times \left(\beta\delta\tau_C^4 + \left(\tau_W\left(\frac{3\tau_W}{\tau} + 4\right)D_{\mathrm{rot}}^2 + 4\beta D_{\mathrm{rot}} + \Omega^2 \tau_W\left(\frac{3\tau_W}{\tau} + 4\right)\right)\tau_C^3\right. \\
& + \left(\frac{\Omega^2 \tau_W^3}{\tau} + 2\Omega^2 \tau_W^2 + D_{\mathrm{rot}}^2\left(\frac{\tau_W}{\tau} + 2\right)\tau_W^2 + 2D_{\mathrm{rot}}\left(\frac{4\tau_W}{\tau} + 5\right)\tau_W + \frac{3\tau_W}{\tau} + 3\right)\tau_C^2 \\
& + \left.\tau_W\left(2D_{\mathrm{rot}}\left(\frac{\tau_W}{\tau} + 2\right)\tau_W + \frac{5\tau_W}{\tau} + 6\right)\tau_C + \tau_W^2\left(\frac{\tau_W}{\tau} + 2\right)\right) \\
& - \frac{2}{\alpha^2 \beta^2 \epsilon^2}\left(\beta\delta\tau_C^4 + \left(\tau_W\left(\frac{3\tau_W}{\tau} + 4\right)D_{\mathrm{rot}}^2 + 4\beta D_{\mathrm{rot}} + \Omega^2 \tau_W\left(\frac{3\tau_W}{\tau} + 4\right)\right)\tau_C^3\right. \\
& + \left(\frac{\Omega^2 \tau_W^3}{\tau} + 2\Omega^2 \tau_W^2 + D_{\mathrm{rot}}^2\left(\frac{\tau_W}{\tau} + 2\right)\tau_W^2 + 2D_{\mathrm{rot}}\left(\frac{4\tau_W}{\tau} + 5\right)\tau_W + \frac{3\tau_W}{\tau} + 3\right)\tau_C^2 \\
& + \left.\tau_W\left(2D_{\mathrm{rot}}\left(\frac{\tau_W}{\tau} + 2\right)\tau_W + \frac{5\tau_W}{\tau} + 6\right)\tau_C + \tau_W^2\left(\frac{\tau_W}{\tau} + 2\right)\right)\left(D_{\mathrm{rot}}\tau_C \tau_W^2 v^2 + \alpha\beta D\epsilon\right) \\
& + \frac{2}{\alpha^2 \beta \epsilon}\left(\tau_C\left(\left(\frac{\tau_W(2D_{\mathrm{rot}}\tau_W + 1)}{\tau} + 1\right)\tau_C^3 + \tau_W\left(2D_{\mathrm{rot}}^2 \tau_W^2 + 2\Omega^2 \tau_W^2 + 3\right)\tau_C^2 + 5D_{\mathrm{rot}}\tau_W^3 \tau_C + \tau_W^3(D_{\mathrm{rot}}\tau_W + 3)\right)v^2\right. \\
& + \alpha D\left(\beta\delta\tau_C^4 + \left(\tau_W\left(\frac{3\tau_W}{\tau} + 4\right)D_{\mathrm{rot}}^2 + 4\beta D_{\mathrm{rot}} + \Omega^2 \tau_W\left(\frac{3\tau_W}{\tau} + 4\right)\right)\tau_C^3\right. \\
& + \left(\frac{\Omega^2 \tau_W^3}{\tau} + 2\Omega^2 \tau_W^2 + D_{\mathrm{rot}}^2\left(\frac{\tau_W}{\tau} + 2\right)\tau_W^2 + 2D_{\mathrm{rot}}\left(\frac{4\tau_W}{\tau} + 5\right)\tau_W + \frac{3\tau_W}{\tau} + 3\right)\tau_C^2 \\
& + \left.\left.\tau_W\left(2D_{\mathrm{rot}}\left(\frac{\tau_W}{\tau} + 2\right)\tau_W + \frac{5\tau_W}{\tau} + 6\right)\tau_C + \tau_W^2\left(\frac{\tau_W}{\tau} + 2\right)\right)\right)\Bigg]
\end{aligned}
$$

**Table 1 | Experimentally measured averaged motility parameters of sperm in different phases**

| Motility pattern (fluid, chemical) | $U$ ($\mu$ms$^{-1}$) | $D_{rot}$ (s$^{-1}$) | $\tau$ (s) | $\Omega$ (s$^{-1}$) | $\tau_C$ (s) | $\tau_W$ (s) |
|---|---|---|---|---|---|---|
| Progressive (Newtonian, not stimulated) | 101 | 0.05 | n.a. | n.a. | n.a. | n.a. |
| PRW (Newtonian, 3 mM caffeine) | 80 | 0.05 | 11.6 | n.a | n.a | n.a |
| PRW (Newtonian, 6 mM caffeine) | 24 | 0.05 | 4.2 | n.a | n.a | n.a |
| Chiral (1% PAM, not stimulated) | 58 | 0.14 | n.a | 0.4 | n.a. | n.a. |
| Chiral (1% PAM, 6 mM caffeine) | 69 | 0.14 | n.a | 1.8 | n.a. | n.a. |
| Circling/wandering (1% PAM, 6 mM caffeine) | 83 | 0.19 | 0.67 | 2.0 | 67 | 35 |

These are used as input for our simulations. 'n.a.' denotes 'not applicable'. These parameter values are rounded according to the experimental error.

with

$$\alpha = \tau_C + \tau_W$$
$$\beta = 1 + \tau^{-1}\tau_W$$
$$\rho = \tau_C - \tau_W + \tau_c\tau_W\tau^{-1}$$
$$\delta = D_{rot}^2 + \Omega^2$$
$$\epsilon = 1 + 2D_{rot}\tau_C + D_{rot}^2\tau_c^2 + \Omega^2\tau_C^2$$
$$\gamma = \tau_C/\tau - 1$$
$$\kappa = 1 - 2D_{rot}\tau_W + D_{rot}^2\tau_W^2 + \Omega^2\tau_W^2$$
$$h = \frac{\alpha\delta}{2}\frac{\left(\tau_C^2\delta + 2D_{rot}\tau_C + 1\right)^2\left(\tau_W^2\delta - 2D_{rot}\tau_W + 1\right)}{\left(\tau_W^2(\tau_C^2\delta + 2D_{rot}\tau_C + 1) - 2\tau_C\tau_W(D_{rot}\tau_C + 1) + \tau_C^2\right)}$$

We obtain the long-time effective diffusivity for the mixed phase $D_{eff}^{CW}$ by taking the limit

$$\lim_{t\to\infty}\frac{\langle|\Delta\vec{r}(t)|^2\rangle}{4t} = D_{eff}^{CW}. \tag{14}$$

The latter is reported in the main text [Eq. (1)].

**Brownian dynamics simulations of active agents in porous media.**
To study the transport of sperm cells in complex geometries, we simulate active agents, following the dynamics described above, in a two-dimensional Lorentz gas, where the obstacles are modeled as overlapping discs of radius $R_0$ in a square domain of length $L$[81]. We ignore hydrodynamic interactions between the swimmers and the obstacles and perform Brownian dynamics simulations. The particles interact with the obstacles via a Weeks-Chandler-Anderson (WCA) potential:

$$\frac{U_i}{k_BT} = \epsilon_{WCA}\begin{cases} 4\left[\left(\frac{R_0}{r_i}\right)^{12} - \left(\frac{R_0}{r_i}\right)^6\right] + 1 & r < 2^{1/6}R_0 \text{ for } i = 1,...N \\ 0, & \text{else}, \end{cases} \tag{15}$$

with the number of obstacles $N$, non-dimensional potential strength $\epsilon_{WCA}$, Boltzmann constant $k_B$, and temperature $T$. Further, the distance between the particle and the $i-$th obstacle is $r_i = |\vec{r}(t) - \vec{r}_i|$, where $\vec{r}_i$ denotes the position of the $i-$th obstacle. The equations of motion are:

$$\frac{d\vec{r}}{dt} = U\vec{e} + \sqrt{2D}\vec{\eta} - \mu\sum_{i=1}^{N}\nabla_i U_i \tag{16a}$$

$$\frac{d\vartheta}{dt} = \Omega + \sqrt{2D_{rot}}\chi, \tag{16b}$$

with mobility $\mu = D/(k_BT)$. Further, $\vec{e} = (\cos\vartheta, \sin\vartheta)$ is the swimming direction and $\vec{\eta}(t)$ and $\chi(t)$ are two independent Gaussian white noise

processes of zero mean and delta-correlated variance, $\langle\eta_i(t)\eta_j(t')\rangle = \delta_{ij}\delta(t-t')$ for $i, j = 1, 2$ and $\langle\chi(t)\chi(t')\rangle = \delta(t-t')$, respectively. We simulate the wandering phase by setting $\Omega = 0$ and changing orientation $\vartheta$ at exponentially distributed times $\sim \exp(-t/\tau)/\tau$. Finally, the mixed phase is simulated by changing from one phase to another at times $\sim \exp(-t/\tau_{C,W})/\tau_{C,W}$; at the switching events, the agent starts moving along a new, random direction. We set the potential strength to $\epsilon_{WCA} = 100$. We further choose parameter values for the active particles, which are extracted from the experimental observations (Table 1) and set the translational diffusivity to $D = 0.01\,\mu\text{m}^2\text{s}^{-1}$.

To study the impact of the porous medium, we vary the radius $R_0 \in [1, 50]\mu m$ of the obstacles by keeping the packing fraction of the obstacles fixed, $\eta = N\pi R_0^2/L^2 = 0.55$ (with $N = 7000$). The latter leads to porous media that have varying mean chord length ($l_c = \pi R_0/(2\eta)$), ranging from $l_c \in [2.86, 143]\mu m$.

For the simulation, we discretize the equations of motion for the chiral phase according to the Euler-Maruyama scheme:

$$\vec{r}(t+\Delta) = \vec{r}(t) + U\vec{e}\Delta - \epsilon_{WCA}\sum_i^N\nabla_i U_i\Delta + \sqrt{2D\Delta}\vec{N}_{\vec{\eta}}(0,1), \tag{17a}$$

$$\vartheta(t+\Delta) = \vartheta(t) + \Omega\Delta + \sqrt{2D_{rot}\Delta}N_\chi(0,1), \tag{17b}$$

where $\vec{N}_{\vec{\eta}}(0,1)$ and $\mathcal{N}_\chi(0,1)$ are independent, normally distributed random variables of zero mean and unit variance. The time step is chosen as $\Delta = 10^{-4}$ and $10^3$ trajectories are obtained up to a total time of $10^4$. The simulation was programmed in Julia using neighbor lists for efficiency[82].

To compute the trapping time distributions $\phi_t(t)$ we measure the instantaneous displacement $\Delta_i^r = |\vec{r}(t_i + \Delta) - \vec{r}(t_i)|$ for 80 trajectories and apply a median filter to smoothen the time-series data. Trapping phases are then classified using the criterion that $\Delta_i^r \leq v\Delta/3$. The long tails of the trapping-time distributions do not change significantly by varying the threshold. We consider a trapping event if its length exceeds $10\Delta$.

## Data availability
The data supporting the findings of this study are available in the article, its Supplementary Information, and the Source Data. Source data are provided with this paper.

## Code availability
The codes used in this study are publicly available at https://doi.org/10.5281/zenodo.18495070.

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

## Acknowledgements
M.Z. thanks Ned S. Wingreen for insightful discussions on sperm circling and wandering dynamics. C.K. acknowledges helpful discussions with S. Mandal and P. Das. M.Z. acknowledges support from the Omenn–Darling Postdoctoral Fellowship, which supported this study. We also acknowledge support from NSF Grant No. DMS/NIGMS 2245850 (to H.A.S. and S.P.).

## Author contributions
M.Z., C.K., and H.A.S. conceived the project. M.Z., Y.L.A., and S.H.C. performed a portion of the experiments, and M.Z. and S.P. carried out the remaining experiments. Y.B., A.P., and C.K. performed the analytical calculations and simulations. M.Z., C.K., and H.A.S. wrote the manuscript, and all authors approved the final version of the manuscript.

## Funding

## Competing interests
The authors declare no competing interests.
