## [Transparent Peer Review file · Nature Communications]

Sperm hyperactivation drives a circling-and-wandering swimming behavior

Corresponding Author: Dr Howard Stone

Version 0:

Reviewer comments:

Reviewer #1

(Remarks to the Author)

In the paper the authors discovered and investigated a novel circling-and-wandering migration strategy of bull sperm cells in the hyperactivation regime in complex fluids with viscoelastic properties. The authors caused hyperactivation by adding 6 mM of caffeine and showed that in a Newtonian fluid the cells motion can be described by stochastic random walk, which they called wandering motion. In a complex fluid with viscoelastic properties the cells followed circular trajectories because of suppression of cells rolling motion. Such behavior the authors called circular motion. However, the authors figured out that in a complex fluid with caffeine stimulation there is a subpopulation of cells which switched between wandering and circling motion. The authors called such motion circling-and-wandering. The authors carefully measured the main parameters of all types of sperm motion: translational speed, persistence time and effective diffusion coefficient. After that the authors experimentally investigated sperm-wall interactions in all the investigated moving regimes showing that in wandering regime the cells scatter from surfaces, while in circling regime they were trapped on pillars with radius 50 μm . Having established experimental parameters of all motion regimes of hyperactivated bull sperm cells the authors put the data into a stochastic model of sperm transport in porous medium. The simulations showed that circling-and-wandering motion provides larger MSD of sperm cells in a porous medium when the average chord length is similar to the size of sperm head, when the progressive sperm remains stuck in the corners. Therefore, the circling-and-wandering motion can be beneficial for sperm cells to navigate in dense porous medium in the hyperactivation regime.

The paper is well written and the provide novel statistically confirmed data about bull sperm cells motility, which might be important for understanding the sperm navigation principles during fertilization process. However, there are several issues, which should be addressed during the revision process.

- 1) Most of the microscopic videos, provided by the authors, contain quick moving white dots, which are significantly smaller than the sperm. The authors should explain the nature of these dots. Are these dots bacteria? In the literature it is well established that bacterial contamination can affect sperm motility. Please provide the data confirming that bacterial contamination hasn't influenced on the obtained results.
- 2) The authors tracked several hundreds of sperm cell in moved in the wandering and the circling motion. However, the authors found only 12 sperm cells of 28 total cells moved in circling-and-wandering motion regime. According to the text they observed such motion only in a single experiment. Please provide statistics on how large is the subpopulation of circling-and-wandering sperm. In how many biological replicates circling-and-wandering sperm cells were observed?
- 3) The authors measured tilt of the sperm head relative to the flagellar beating plane which was 13.0 ± 3.5 degrees for all the cells. However, according to the previous results (doi: 10.1073/pnas.1515159112) all human sperm cells can be divided into two subpopulations according to the tilt angle. Could the authors provide the data about the tilt angle of cells that can switch from circling to wandering motion in the hyperactivation regime?

Reviewer #2

(Remarks to the Author)

This manuscript presents an experimental characterization of bull sperm hyperactivation in Newtonian and complex fluids near surfaces, designed to mimic the complex biological environment of the reproductive tract. The observation of a circling-and-wandering mode, where motility alternates between circling and wandering periods, is particularly intriguing. The investigations of sperm-wall interactions during different motility phases near concave and convex boundaries are also

valuable. Taken together, the findings suggest that this circling-and-wandering migration strategy outperforms either mode alone in spreading through porous media. The study is well executed and thoroughly documented, and the findings are both interesting and impactful. I recommend publication and have only two comments for the authors to consider.

First, while I understand that circling motility observed in complex fluids and its connection to the rolling motion of the sperm has been discussed in earlier studies, it may still be helpful to briefly discuss the physical mechanisms underlying distinct motility modes. This may help provide readers with a clearer framework for understanding the transition between these modes.

Second, the circling-and-wandering motility is a compelling finding, and the authors provide a reasonable argument that this behavior is intrinsic to hyperactivation rather than a consequence of rheological heterogeneity in the medium. This argument would be further strengthened by additional evidence quantifying the extent of heterogeneity, thereby excluding it as a significant factor. For future work, I am also curious whether circling-and-wandering motility occurs more generally in other viscoelastic fluids or in actual reproductive tract mucus.

Reviewer #3

(Remarks to the Author)

This manuscript reports sperm moving in a Newtonian fluid with and without circular pillars. Experiments show three swimming gaits: wandering, circling, and a new circling-and-wandering mode that consists of stochastic transitions between the two. Introducing pillars results in sperm scattering and trapping depending on the swimming gait and the pillar's curvature. Stochastic simulations in porous media suggest that circling-and-wandering exhibits higher effective diffusivity than wandering or circling alone.

The manuscript's noteworthy results are in observing the curious transitions between wandering and circling, the theoretical prediction for the effective diffusivity based on renewal theory, and the simulations showing non-trivial diffusion in porous media. The work appears related to micro swimmers scattering and trapping around an obstacle [1] and transitioning between wandering and circling [2]. The novelty of the current study and how it compares with the established literature on micro swimmers warrant clarification.

The conclusion that the circling-and-wandering mode has the highest effective diffusivity in porous media is interesting but only supported by stochastic simulations. The simulations represent each sperm as a particle with orientation that is assumed to change randomly each time it transitions between wandering and circling. This assumption is questionable and not supported by experiments or theory. Additional evidence based on experiments and theory would strengthen the conclusion.

The renewal theory seems to rely on the transitions between wandering and circling phases occurring at exponentially distributed times. However, experimental evidence for this is lacking. Only the average times spent in each phase is reported with a small sample size of $N=12$.

Claims about migration in the title, abstract, introduction and discussion are premature. The current study shows only the mean square displacement and the effective diffusivity under different conditions. Figure 6 seems to show a gradient in hyperactivation level but the message was unclear. The results do not immediately translate to sperm migration.

The methodology is sound and there is enough detail provided in the methods for the work to be reproduced.

References

[1] Spagnolie, S. E., Moreno-Flores, G. R., Bartolo, D., & Lauga, E. (2015). Geometric capture and escape of a microswimmer colliding with an obstacle. *Soft Matter*

[2] Takagi, D., Braunschweig, A. B., Zhang, J., & Shelley, M. J. (2013). Dispersion of self-propelled rods undergoing fluctuation-driven flips. *Phys. Rev. Lett.*

Version 1:

Reviewer comments:

Reviewer #1

(Remarks to the Author)

During the revision process the authors answered all my and other reviewers questions, performing additional experiments and data analysis, which allowed them to significantly improved the manuscript. Now all the claims are confirmed by the data, the methodology is adequate, and well described. Therefore, the manuscript might be accepted for publication.

Reviewer #2

(Remarks to the Author)

The authors have satisfactorily addressed the questions and comments in my previous report. I appreciate the additional experiments performed to address the possibility that circling-and-wandering motility arises from spatial heterogeneity in the rheological properties of the medium. I recommend publication of this work.

Reviewer #3

(Remarks to the Author)

The authors have adequately addressed all my concerns. I recommend proceeding with publication.

Reply to Referee 1

We thank the referee for taking the time to read the manuscript and for the constructive criticism. We provide a blue-line version of our manuscript to highlight our changes.

Referee 1: In the paper the authors discovered and investigated a novel circling-and-wandering migration strategy of bull sperm cells in the hyperactivation regime in complex fluids with viscoelastic properties. The authors caused hyperactivation by adding 6 mM of caffeine and showed that in a Newtonian fluid the cells motion can be described by stochastic random walk, which they called wandering motion. In a complex fluid with viscoelastic properties the cells followed circular trajectories because of suppression of cells rolling motion. Such behavior the authors called circular motion. However, the authors figured out that in a complex fluid with caffeine stimulation there is a subpopulation of cells which switched between wondering and circling motion. The authors called such motion circling-and-wandering. The authors carefully measured the main parameters of all types of sperm motion: translational speed, persistence time and effective diffusion coefficient. After that the authors experimentally investigated sperm-wall interactions in all the investigated moving regimes showing that in wandering regime the cells scatter from surfaces, while in circling regime they were trapped on pillars with radius 50 μm . Having established experimental parameters of all motion regimes of hyperactivated bull sperm cells the authors put the data into a stochastic model of sperm transport in porous medium. The simulations showed that circling-and-wandering motion provides larger MSD of sperm cells in a porous medium when the average chord length is similar to the size of sperm head, when the progressive sperm remains stuck in the corners. Therefore, the circling-and-wandering motion can be beneficial for sperm cells to navigate in dense porous medium in the hyperactivation regime.

The paper is well written and the provide novel statistically confirmed data about bull sperm cells motility, which might be important for understanding the sperm navigation principles during fertilization process. However, there are several issues, which should be addressed during the revision process.

Reply: We thank the referee for stating that our work is “*well written*” and provides “*novel statistically confirmed data about bull sperm cells motility*”. We gladly address the referee’s comments below.

Referee 1: 1) Most of the microscopic videos, provided by the authors, contain quick moving white dots, which are significantly smaller than the sperm. The authors should explain the nature of these dots. Are these dots bacteria? In the literature it is well established that bacterial contamination can affect sperm motility. Please provide the data confirming that bacterial contamination hasn’t influenced on the obtained results.

Reply: We thank the reviewer for this comment. To address this point, we performed new experiments in which all sperm preparation steps were carried out under sterile conditions in a biosafety cabinet. The resulting measurements are consistent with our previously reported results, independent of the nature of the fast-moving dots observed in the videos. Based on these new experiments, we have updated Supplementary Videos 2, 3, and 4. In these new videos, the round-shaped dots likely originate from components of the egg yolk extender used for sperm preservation, such as microscopic lipid- or protein-rich particles, although we do not assign a specific identity to them. The swimming behavior space shown in Fig. 1F, which is now obtained from 5 independent sperm samples, has also been updated using the newly obtained results. In Figure R1, we show the original dataset, the additional dataset collected during revision, a plot comparing these datasets, and the updated swimming behavior space, which is now included in the revised manuscript.

In addition, we performed new experiments to check the consistency of our observations of circling-and-wandering behavior. The revised manuscript now reports 18 tracks exhibiting circling-and-wandering behavior from a total of 48 cells, collected across 5 independent samples. Figures 2D, S3, and S4 have been updated accordingly using these newly obtained trajectories.

Finally, because the new measurements led to updated parameters in Table 1, we repeated all simulations and updated Fig. 5 accordingly. The simulation outcomes remain consistent with those reported in the original version of the manuscript. Overall, these new experiments demonstrate that the quantitative claims of the paper are independent of the nature of the dots that were present in the sperm samples in the original version of the manuscript. Moreover, the new experiments enriched our data analysis and expanded the statistical basis of the results.

Figure R 1: Original datasets and additional datasets collected during revisions for the swimming behavior space (top), as well as comparison and combined datasets for the behavior space (bottom).

Referee 1: 2) The authors tracked several hundreds of sperm cell in moved in the wandering and the circling motion. However, the authors found only 12 sperm cells of 28 total cells moved in circling-and-wandering motion regime. According to the text they observed such motion only in a single experiment. Please provide statistics on how large is the subpopulation of circling-and-wandering sperm. In how many biological replicates circling-and-wandering sperm cells were observed?

Reply: We thank the reviewer for this helpful comment.

To provide a statistical context for circling-and-wandering motility, we first quantified the fraction of sperm cells that resumed rolling after initial suppression, as rolling resumption serves as a physical indicator of the transition from circling to wandering. In TALP + 1% PAM solution and under 6 mM caffeine treatment, $91 \pm 5\%$ of cells had their rolling initially suppressed upon entering the chamber, and $27 \pm 4\%$ of these cells subsequently resumed rolling (Figure S2). These measurements were collected from three different sperm samples, each observed across 3–4 independent chambers (Table R1), providing an estimate of the fraction of cells with circling-and-wandering motility.

To accurately observe this motility pattern, we performed experiments at extremely low sperm densities for two main reasons. First, this ensured that circling-and-wandering behavior was not influenced by interactions between sperm or by the collective impact of the medium’s rheological properties. Second, low densities allowed us to track individual cells for extended periods, which is extremely challenging at high densities. This approach, however, comes with limitations, primarily a relatively small sample size. In the Discussion section, we motivate future studies aimed at developing techniques to observe circling-and-wandering motility more efficiently and at higher throughput.

Regarding the circling-and-wandering motility, in our original dataset this behavior was observed across three independent sperm samples (Tables R2–R4). During the revision process, we collected data from two additional biological samples (Tables R5–R6). The detailed statistics for circling-and-wandering cells across all five samples are provided in Tables R2–R6, showing that this motility pattern is consistently observed across multiple biological replicates. We have now revised the manuscript accordingly, and Tables R1–R6 are included in the Supplementary Information as Tables S1–S6.

	Sample 1			Sample 2			Sample 3			
Chamber ID	1	2	3	1	2	3	1	2	3	4
Number of cells	57	38	49	31	24	38	50	41	59	34
Number of cells with rolling suppressed	53	34	43	29	20	34	48	34	55	33
Number of cells with resumed rolling	14	10	13	7	4	10	11	11	16	10

Table R1: Statistics of suppressed and resumed rolling.

Chamber ID	1	2	3	4	5	6	7	8	Total
Number of cells in the chamber	1	2	1	1	2	1	1	1	10
Number of cells with C&W	0	1	0	1	0	1	0	0	3

Table R2: Statistics of circling and wandering in sample 1.

Chamber ID	1	2	3	4	5	6	Total
Number of cells in the chamber	2	1	2	1	1	1	8
Number of cells with C&W	0	1	1	1	0	0	3

Table R3: Statistics of circling and wandering in sample 2.

Chamber ID	1	2	3	4	5	6	7	Total
Number of cells in the chamber	1	2	2	1	2	1	1	10
Number of cells with C&W	1	0	2	1	0	1	1	6

Table R4: Statistics of circling and wandering in sample 3.

Chamber ID	1	2	3	4	5	6	Total
Number of cells in the chamber	1	2	2	2	1	2	10
Number of cells with C&W	0	1	1	0	0	0	2

Table R5: Statistics of circling and wandering in sample 4.

Chamber ID	1	2	3	4	5	6	7	8	Total
Number of cells in the chamber	1	1	2	1	2	1	1	1	10
Number of cells with C&W	1	0	0	1	1	1	0	0	4

Table R6: Statistics of circling and wandering in sample 5.

Referee 1: 3) The authors measured tilt of the sperm head relative to the flagellar beating plane which was 13.0 ± 3.5 degrees for all the cells. However, according to the previous results (doi: 10.1073/pnas.1515159112) all human sperm cells can be divided into two subpopulations according to the tilt angle. Could the authors provide the data about the tilt angle of cells that can switch from circling to wandering motion in the hyperactivation regime?

Reply: We thank the reviewer for this insightful comment. In response, we have measured the sperm head tilt for different motility modes and now provide the corresponding statistics in Figure S6. Specifically, we quantified the head angle during pure wandering, during the wandering periods of circling-and-wandering trajectories, and the initial head angle of cells that later exhibited either pure circling or circling-and-wandering motility. Our analysis shows that the head tilt for pure wandering is $\varphi_W = 13 \pm 2^\circ$, while during the wandering periods of circling-and-wandering trajectories it is $\varphi_{CW} = 12 \pm 2^\circ$ ($N = 5$), a difference that is not statistically significant ($p \approx 0.41$, Fig. S6a). Similarly, the initial head angle of cells that later exhibited pure circling was $\varphi_C^{\text{initial}} = 12 \pm 4^\circ$, compared to $\varphi_{CW}^{\text{initial}} = 12 \pm 2^\circ$ for cells that later exhibited circling-and-wandering ($p \approx 0.83$, Fig. S6b).

These results indicate that the head tilt is largely conserved across all motility patterns that involve rolling, suggesting that the observed circling-and-wandering behavior is primarily dictated by the sperm beating pattern and the medium's rheology rather than by intrinsic differences in head tilt.

This analysis is reflected on page 6 of the revised manuscript.

Reply to Referee 2

We thank the referee for taking the time to read the manuscript and for the constructive criticism. We provide a blue-line version of our manuscript to highlight our changes.

Referee 2: This manuscript presents an experimental characterization of bull sperm hyperactivation in Newtonian and complex fluids near surfaces, designed to mimic the complex biological environment of the reproductive tract. The observation of a circling-and-wandering mode, where motility alternates between circling and wandering periods, is particularly intriguing. The investigations of sperm-wall interactions during different motility phases near concave and convex boundaries are also valuable. Taken together, the findings suggest that this circling-and-wandering migration strategy outperforms either mode alone in spreading through porous media. The study is well executed and thoroughly documented, and the findings are both interesting and impactful. I recommend publication and have only two comments for the authors to consider.

Reply: We are grateful that the referee recommends publication after addressing two remarks. We are also glad that the referee finds our work “*well executed and thoroughly documented*” and “*interesting and impactful*”. The referee’s comments are addressed below.

Referee 2: First, while I understand that circling motility observed in complex fluids and its connection to the rolling motion of the sperm has been discussed in earlier studies, it may still be helpful to briefly discuss the physical mechanisms underlying distinct motility modes. This may help provide readers with a clearer framework for understanding the transition between these modes.

Reply: We thank the reviewer for this comment. To address it and provide a clearer framework for understanding the transition between these modes, we revised the manuscript on page 3 to briefly discuss the physical mechanisms underlying both circling and wandering motions. Additionally, on pages 14 and 15 in the Discussion section, we note that while we have characterized circling-and-wandering motion, we emphasize the need for future studies to investigate what actually **drives** the transitions between circling and wandering.

In addition to these essential revisions, we also measured the sperm head tilt angle relative to the flagellar beating plane during the wandering periods of circling-and-wandering in the complex fluid and compared it to the tilt angle during pure wandering in Newtonian fluid; the corresponding statistics are presented in Figure S6. Our analysis shows that the head tilt for pure wandering is $\varphi_W = 13 \pm 2^\circ$, while during the wandering periods of circling-and-wandering trajectories it is $\varphi_{CW} = 12 \pm 2^\circ$ ($N = 5$), a difference that is not statistically significant ($p \approx 0.41$, Fig. S6a). Similarly, the initial head angle of cells that later exhibited pure circling was $\varphi_C^{\text{initial}} = 12 \pm 4^\circ$, compared to $\varphi_{CW}^{\text{initial}} = 12 \pm 2^\circ$ for cells that later exhibited circling-and-wandering ($p \approx 0.83$, Fig. S6b).

We believe this new experimental measurement will be valuable for understanding the physical mechanisms underlying wandering motion in both complex and Newtonian fluids. It indicates that the generic features of wandering motion are determined by asymmetry in the flagellar beating plane as well as the head tilt angle, and it also motivates future studies on the head tilt during the circling phase, which we discuss on page 14.

Referee 2: Second, the circling-and-wandering motility is a compelling finding, and the authors provide a reasonable argument that this behavior is intrinsic to hyperactivation rather than a consequence of rheological heterogeneity in the medium. This argument would be further strengthened by additional evidence quantifying the extent of heterogeneity, thereby excluding it as a significant factor. For future work, I am also curious whether circling-and-wandering motility occurs more generally in other viscoelastic fluids or in actual reproductive tract mucus.

Reply: We thank the referee for this insightful remark.

To address the possibility that circling-and-wandering motility arises from spatial heterogeneity in the rheological properties of the medium, we performed additional microrheological experiments to directly quantify the effective viscosity of the fluid. Specifically, we added fluorescent polystyrene microspheres (1 μm diameter) to TALP+1% PAM and tracked their motion for five minutes at 10 FPS across multiple chamber locations (Fig. R2a).

From eight regions and more than one hundred microspheres per region, the mean MSDs collapsed and showed consistent long-timescale diffusive behavior (Fig. R2b). The resulting effective diffusivity was $D_{\text{eff}} = (7.5 \pm 0.5) \times 10^{-3} \mu\text{m}^2\text{s}^{-1}$, corresponding to an effective viscosity of $60 \pm 4 \text{ mPa}\cdot\text{s}$ (Fig. R2c). We note that the error bar lies below the uncertainty in our measurements of both diffusivity and viscosity within

each region. This viscosity range is orders of magnitude higher than that of TALP (≈ 1 mPa s), where pure wandering occurs.

These findings indicate that the emergence of circling-and-wandering reflects an intrinsic feature of hyperactivated motion in a complex fluid, rather than an artifact of sharply separated Newtonian and non-Newtonian domains. Overall, our results suggest that a potentially more complex mechanochemical phenomenon drives the transitions between circling and wandering motion. We have revised the manuscript on pages 6 and 7 and added Figure S7 to the Supplementary Information file. In line with the referee's comment, we have also revised the Discussion section on page 14 and highlighted the need for future studies in more physiologically representative environments, particularly the mucus of the female reproductive tract.

Figure R 2: **Microrheological measurement in TALP + 1% PAM.** a) Microspheres with an average diameter of $1 \mu\text{m}$ were added to the solution, and their motion was recorded in eight regions of the chamber. b) MSDs corresponding to tracks acquired at 10 FPS over five minute intervals from more than 100 microspheres in each region. All regions show diffusive behavior at long times. c) Average MSDs for each region collapse to a single master curve.

Reply to Referee 3

We thank the referee for taking the time to read the manuscript and for the constructive criticism. We provide a blue-line version of our manuscript to highlight our changes.

Referee 3: This manuscript reports sperm moving in a Newtonian fluid with and without circular pillars. Experiments show three swimming gaits: wandering, circling, and a new circling-and-wandering mode that consists of stochastic transitions between the two. Introducing pillars results in sperm scattering and trapping depending on the swimming gait and the pillar’s curvature. Stochastic simulations in porous media suggest that circling-and-wandering exhibits higher effective diffusivity than wandering or circling alone.

The manuscript’s noteworthy results are in observing the curious transitions between wandering and circling, the theoretical prediction for the effective diffusivity based on renewal theory, and the simulations showing non-trivial diffusion in porous media. The work appears related to micro swimmers scattering and trapping around an obstacle [1] and transitioning between wandering and circling [2]. The novelty of the current study and how it compares with the established literature on micro swimmers warrant clarification.

References

- [1] Spagnolie, S. E., Moreno-Flores, G. R., Bartolo, D., & Lauga, E. (2015). Geometric capture and escape of a microswimmer colliding with an obstacle. *Soft Matter*
- [2] Takagi, D., Braunschweig, A. B., Zhang, J., & Shelley, M. J. (2013). Dispersion of self-propelled rods undergoing fluctuation-driven flips. *Phys. Rev. Lett.*

Reply: We thank the referee for acknowledging our observations and theoretical results concerning the circling-and-wandering motility pattern as ‘*noteworthy*’. We gladly comment on the references provided by the referee in the context of our work:

First, in their theoretical work Spagnolie et al. [1] discuss the interactions of a microswimmer – modeled as a point force-dipole in the far-field – near a spherical obstacle. They reveal that swimmers scatter away from small obstacles and can be captured by and orbit around larger obstacles, which depends on the orientation of the microswimmer relative to the obstacle and the dipole strength. While this is an important model system, which can capture qualitatively the behavior of colloids near obstacles, our experimental system – bull sperm – are much more complex. Our observations show that the sperm-obstacle interactions involve the sperm head, its beating tail, the obstacle, and the confinement of the set-up, which can lead to intriguing phenomena, such as reversals. So, while in principle sperm could be modeled as a force-dipole in the far field, their near-field flows are much more complex and unraveling their importance, relative to steric effects, for sperm-obstacle interactions remains an open research endeavor.

Second, the experimental work by Takagi et al. [2] deals with the motion of self-propelled catalytic rods, which tend to move along large circles and change stochastically their sense of rotation. This effect is attributed to the flipping of slightly-curved rods and results from a purely mechanical interaction between the rod and the surface. We note that the relevance of this finding for our work is not very clear to us. Nevertheless, we want to first stress that circular near-surface motion can emerge in any active systems with small asymmetries and has been reported in various microorganisms, including sperm, before. Moreover, the switching between phases in our system is most likely driven by an internal biophysical machinery that induces different flagella beating patterns, rather than by mechanical interactions alone.

To summarize, while the two papers cited by the referee are related to microswimmers and their obstacle interactions/motility patterns, our experimental system of hyperactivated bull sperm is very different and shows distinct new phenomena, which could be highly relevant in the context of fertilization.

In response to the referee’s remark, we have now added the following paragraph to our conclusions:

“The circling dynamics of sperm, quantified in this work, are inherent to a large class of active agents near boundaries, including bacteria [66], where the rotation of the flagella bundle and the counter-rotation of the cell body lead to circular motion as result of hydrodynamic interactions with the wall, and self-propelled colloids [67,68], whose shape-asymmetries introduce circular dynamics. Despite geometrical effects, circular motion has also been identified in spherical, achiral Janus colloids immersed in a non-Newtonian medium[69], whose angular velocity can switch sign. In our work – in addition to geometry and rheology – the angular velocity strongly depends on the amount of caffeine and, consequently, a more complex interplay of biophysical

and mechanical features appears to be important that needs to be unraveled in the future. Due to these intricate dynamics, sperm interactions with discs differ from previous experiments of self-propelled rods [70] and theoretical work on microswimmers [71], where the interplay of noise and hydrodynamic interactions govern the detention times at the boundaries. In particular, how steric interactions of the long sperm flagellum together with these hydro-chemical effects result in trapping by obstacles represents an interesting research question.”

Referee 3: The conclusion that the circling-and-wandering mode has the highest effective diffusivity in porous media is interesting but only supported by stochastic simulations. The simulations represent each sperm as a particle with orientation that is assumed to change randomly each time it transitions between wandering and circling. This assumption is questionable and not supported by experiments or theory. Additional evidence based on experiments and theory would strengthen the conclusion.

Reply: We thank the referee for this insightful comment. Extracting reliably the angle after the change from one phase to another from experimental measurements is very challenging as these transitions are hard to detect precisely and so the estimates are noisy. Thus, to address the referee’s remark, we have performed additional simulations, in which the agent’s swimming direction does not change at the transition between both phases. Our results barely change for the long-time effective diffusivity for all packing fractions considered, see Fig. R3. In particular, the relative error amounts to less than 0.1%. This is because both phases – circling and wandering – already lead to substantial changes of the agent’s swimming direction at smaller time scales ($1/\Omega \approx 0.5\text{s} \lesssim \tau_C \approx 67\text{s}$ and $\tau \approx 0.7\text{s} \lesssim \tau_W \approx 35\text{s}$) and hence one additional change of swimming direction at the renewal event does not modify the results.

Figure R 3: Long-time effective diffusivities D_{eff} . (Left panel) The yellow symbols correspond to the results for the mixed phase, where the agent randomly changes its swimming direction between two phases, and the dark purple symbols show the results for agents that do not change swimming direction at each phase change. (Right panel) Relative difference between the two models for the mixed phase. D_{eff} corresponds to a random change of orientation at the transition between phases and $D_{\text{eff}}^{\text{w/o}}$ corresponds to no change in orientation.

We have added a note to our conclusions and included Fig. R3 in Supplementary Information as Fig. S14:

“Another assumption of our theory is that sperm randomly change their swimming direction at the transition between two phases. This is reasonable because both phases – circling and wandering – already lead to substantial changes of the agent’s swimming direction at smaller time scales ($1/\Omega \approx 0.5\text{s} \lesssim \tau_C \approx 70\text{s}$ and $\tau \approx 0.7\text{s} \lesssim \tau_W \approx 40\text{s}$) and hence one additional change of swimming direction at the transition does not contribute to the results (Fig. S14).”

Referee 3: The renewal theory seems to rely on the transitions between wandering and circling phases occurring at exponentially distributed times. However, experimental evidence for this is lacking. Only the average times spent in each phase is reported with a small sample size of $N=12$.

Reply: We thank the referee for raising this point. While we have increased our sample size to $N = 18$ with our new experiments, our data points (now included in Supplementary Information as Fig. S2b) are not sufficient for estimating the distribution of circling and wandering times reliably.

We, however, believe that an exponential distribution represents a reasonable starting point, as any other distribution would involve additional parameters, which may be hard/impossible to extract reliably from our data. This choice of distribution is substantiated by the nice agreement between the mean-square displacements of experiments and theory (in Fig. 2d of our manuscript), which does not involve any fitting parameter. We would like to point out that the theory can be readily applied for arbitrary distributions for the circling and wandering times (that have a finite mean) and thus serves as starting point for future studies on circling-and-wandering dynamics that involve higher-order moments or the full probability distributions, which can provide additional information regarding the choice of distributions.

We reflected this response in the manuscript by adding a note to our conclusions:

“It is important to note that our theoretical predictions rely on the choice of exponential distributions for the circling and wandering times, which nicely described our data for the mean-square displacements without any fitting parameters. Our theory can, however, be readily applied for arbitrary distributions that have a finite mean. Thus, future work may focus on deriving higher-order moments for the displacements or the intermediate scattering functions (i.e. the Fourier transform of the probability densities) [56,65], providing access to the full spatiotemporal dynamics. The latter may enable insights into the role of the distributions for sperm dynamics.”

Additionally, we appreciate the reviewer’s concern and acknowledge that the statistical power of our analysis of circling-and-wandering dynamics is limited. Despite extensive experimental effort, our current approach yielded 18 hyperactivated tracks out of 48. Accordingly, on page 14 of the Discussion section, we motivate future studies aimed at developing new experimental methods that can visualize and quantify the motion of sufficiently large sperm populations in complex fluids over timescales of minutes to hours. Such studies will enable a more detailed characterization of how circling-and-wandering dynamics unfold over time.

Referee 3: Claims about migration in the title, abstract, introduction and discussion are premature. The current study shows only the mean square displacement and the effective diffusivity under different conditions.

Reply: We thank the reviewer for this remark. We have revised the title, introduction, and discussion of our manuscript to avoid potential confusion. In particular, we have changed “migration strategy” to “swimming behavior” in the title of our manuscript.

Referee 3: Figure 6 seems to show a gradient in hyperactivation level but the message was unclear. The results do not immediately translate to sperm migration.

Reply: We have also removed the gradient in Fig. 6 and revised the caption to highlight how our results may have potential implications for sperm migration in the FRT.

Referee 3: The methodology is sound and there is enough detail provided in the methods for the work to be reproduced.

Reply: Thank you!